# Overlay databank unlocks data-driven analyses of biomolecules for all

Anne M. Kiirikki[1], Hanne S. Antila[2,3], Lara S. Bort[2,4], Pavel Buslaev[5], Fernando Favela-Rosales [6], Tiago Mendes Ferreira[7], Patrick F. J. Fuchs [8,9], Rebeca Garcia-Fandino[10], Ivan Gushchin, Batuhan Kav [11,12], Norbert Kučerka[13], Patrik Kula[14], Milla Kurki [15], Alexander Kuzmin, Anusha Lalitha [16], Fabio Lolicato [17,18], Jesper J. Madsen [19,20], Markus S. Miettinen [2,21,22], Cedric Mingham[23], Luca Monticelli [24,25], Ricky Nencini[1,26], Alexey M. Nesterenko[21,22], Thomas J. Piggot[27], Ángel Piñeiro[28], Nathalie Reuter[21,22], Suman Samantray [11,29], Fabián Suárez-Lestón [10,28,30], Reza Talandashti[21,22] & O. H. Samuli Ollila [1,31] ✉

Tools based on artificial intelligence (AI) are currently revolutionising many fields, yet their applications are often limited by the lack of suitable training data in programmatically accessible format. Here we propose an effective solution to make data scattered in various locations and formats accessible for data-driven and machine learning applications using the overlay databank format. To demonstrate the practical relevance of such approach, we present the NMRlipids Databank—a community-driven, open-for-all database featuring programmatic access to quality-evaluated atom-resolution molecular dynamics simulations of cellular membranes. Cellular membrane lipid composition is implicated in diseases and controls major biological functions, but membranes are difficult to study experimentally due to their intrinsic disorder and complex phase behaviour. While MD simulations have been useful in understanding membrane systems, they require significant computational resources and often suffer from inaccuracies in model parameters. Here, we demonstrate how programmable interface for flexible implementation of data-driven and machine learning applications, and rapid access to simulation data through a graphical user interface, unlock possibilities beyond current MD simulation and experimental studies to understand cellular membranes. The proposed overlay databank concept can be further applied to other biomolecules, as well as in other fields where similar barriers hinder the AI revolution.

Tools based on artificial intelligence (AI) are currently revolutionising many fields, yet their success relies on training data that is available in programmatically accessible and standardised format[1]. For example in structural biology, Protein Data Bank[2] has enabled revolutionary tools that can predict protein structures with unprecedented accuracy[3]. However, the lack of smart guidelines and community-consensus best-practices for data sharing are limiting the development of such databanks and AI applications in many fields[1]. This is the case in biomolecular modelling, where vast amount of data is already available for proteins, lipids, nucleic acids, and carbohydrates, but tools that enable applications of these data for data-driven applications are not yet available[4]. Here we propose a cost-effective solution to make data

scattered in various locations and formats accessible for data-driven and machine learning (ML) applications: an overlay databank. We demonstrate the practical relevance of such approach for understanding cellular membranes by incorporating available lipid bilayer simulations into the NMRlipids Databank. Importantly, the basic principles of an overlay databank can be applied to simulation data of any other biomolecules (that are becoming available in increasing amounts[4]), as well as to any other field where the development of AI-based tools is limited by the lack of data access and community best-practices—such as the assignment of NMR spectra[5]. Incentives to share data can be further accelerated by combining overlay databanks with an open collaboration approach (see ref. [6]).

Cellular membranes contain hundreds of different types of lipid molecules that regulate the membrane properties, morphology, and biological functions[7–9]. Membrane lipid composition is implicated in diseases, such as cancer and neurodegenerative disorders, and therapeutics that affect membrane compositions are emerging[10]. However, biomembranes are often difficult to study experimentally, because they are complex mixtures of proteins and lipids in disordered fluid state with complicated phase behaviour at biological conditions. Molecular dynamics (MD) simulations can be used to model biomembranes in detail, but the computational cost for simulating all possible biological membrane compositions would be formidable. For those reasons, data-driven and machine-learning-based models that predict biomembrane properties will benefit wide range of fields covering academia and industry, from cell membrane biology to lipid nanoparticle formulations.

Here we present the NMRlipids Databank—a community-driven, open-for-all database featuring programmatic access to atom-resolution MD simulations of lipid bilayers. To demonstrate its advantages over existing approaches, we build a ML model that predicts membrane properties from its lipid composition and show how information on rare phenomena that are beyond the scope of standard MD simulation investigations can be gleaned from the Databank. In addition, we demonstrate the immediate relevance of NMRlipids Databank in extending the scope of MD simulations to new fields: Using a data-driven approach, we are able to analyse how anisotropic diffusion of water depends on membrane properties; this benefits understanding in magnetic resonance imaging (MRI)[11] and pharmacokinetics[12], where MD simulations have, until now, been rarely applied. Furthermore, the Databank performs automatic quality evaluation of membrane simulations, which facilitates the selection of best-performing models for each given application and accelerates the development of simulation parameters and methodology. Notably, the overlay-databank and open-collaboration approaches would facilitate the collecting of community-contributed data and providing programmatic access to them also in fields other than membrane simulations. For example, substantial biomolecular MD simulation data are already available but in scattered locations and formats[4]. The technical advances presented here will immediately benefit making these data programmatically accessible. Furthermore, the overlay databank configuration will benefit a broad range of fields where access to training data is a bottleneck for building AI-based tools.

## Results

### NMRlipids overlay databank delivers access to MD simulations of membranes composed of the biologically most abundant lipids

NMRlipids Databank is a community-driven catalogue containing atomistic MD simulations of biologically relevant lipid membranes emerging from the NMRlipids open collaboration[6,13–16]. It has been designed to improve the Findability, Accessibility, Interoperability, and Reuse[17] of MD simulation data, most importantly the output trajectories and necessary information to their reuse. The NMRlipids Databank is constructed using the NMRlipids project protocol, in

which all the content is openly accessible throughout the project[6]. Currently, the NMRlipids Databank contains 765 simulation trajectories with the total length of approximately 0.4 ms. Single-component lipid membranes and binary mixtures are currently most abundant in the NMRlipids Databank, yet mixtures with up to five lipid types are available. For available mixtures, see Fig. 1E. The distribution of lipids among the available simulations, shown in Fig. 1B, roughly resembles the biological relative abundance of different lipid types, with phosphatidylcholine (PC) being the most common followed by cholesterol, phosphatidylethanolamine (PE), phosphatidylserine (PS), phosphatidylglycerol (PG), phosphatidylinositol (PI), and other lipids, depending on organism and organelle[7]. Abbreviations and full names of all lipids present in the Databank are listed in the NMRlipids Databank documentation[18]. Force fields used in simulations cover all the essential parameter sets commonly used in lipid simulations, see Fig. 1C and Supplementary Table 1, including also united atom and polarisable force fields. Therefore, the averages calculated over the Databank can be considered as mean predictions from available lipid models (average over force field parameters) for an average cell membrane (average over lipid compositions).

The overlay structure of the NMRlipids Databank, illustrated in Fig. 1A, is designed to enable efficient upcycling of MD simulations for data-driven and ML applications with minimal investment on new infrastructure. Raw simulation data in the *Data layer* can be stored in any publicly available location with long term stability and with permanent links to the data, such as digital object identifiers (DOIs), such as Zenodo (zenodo.org). The *Databank layer* (github.com/NMRlipids/Databank) is the core of the Databank containing all the relevant information about the simulations: links to the raw data, relevant metadata describing the systems, universal naming conventions for lipids and their atoms, quality evaluation of simulations against experimental data, and the computer programmes to create the entries and to analyse the five basic properties extracted from all simulations (area per lipid, C–H bond order parameters, X-ray scattering form factors, membrane thickness, and equilibration times of principal components). Also the values for these five basic properties are stored in the *Databank layer*.

The *Application layer* is composed of repositories and tools that read information from the *Databank layer* for further analyses. Because the *Application layer* does not interfere with the *Databank layer*, it can be freely extended by anyone for a wide range of purposes. This is demonstrated here with two examples: the NMRlipids Databank graphical user interface (NMRlipids Databank-GUI) at databank. nmrlipids.fi and a repository exemplifying novel analyses utilising NMRlipids Databank as discussed below (github.com/NMRlipids/ DataBankManuscript). A more detailed description of the NMRlipids Databank structure is available in the Supplementary Information.

### NMRlipids Databank-GUI: graphical access to the MD simulation data

NMRlipids Databank-GUI, available at databank.nmrlipids.fi, provides easy access to the NMRlipids Databank content through a graphical user interface (GUI). Simulations can be searched based on their molecular composition, force field, temperature, membrane properties, and quality; the search results are ranked based on the simulation quality as evaluated against experimental data when available. Membranes can be visualised, and properties between different simulations and experiments compared. The NMRlipids Databank-GUI enables rapid surveying of what simulation data is available, selection of the best available simulations for specific systems based on ranking lists, and comparisons of basic properties between different types of membranes. Notably, the GUI enables these operations to be performed by scientists with a wide range of backgrounds—including those who do not necessarily have programming expertise or other means to access MD simulation data.

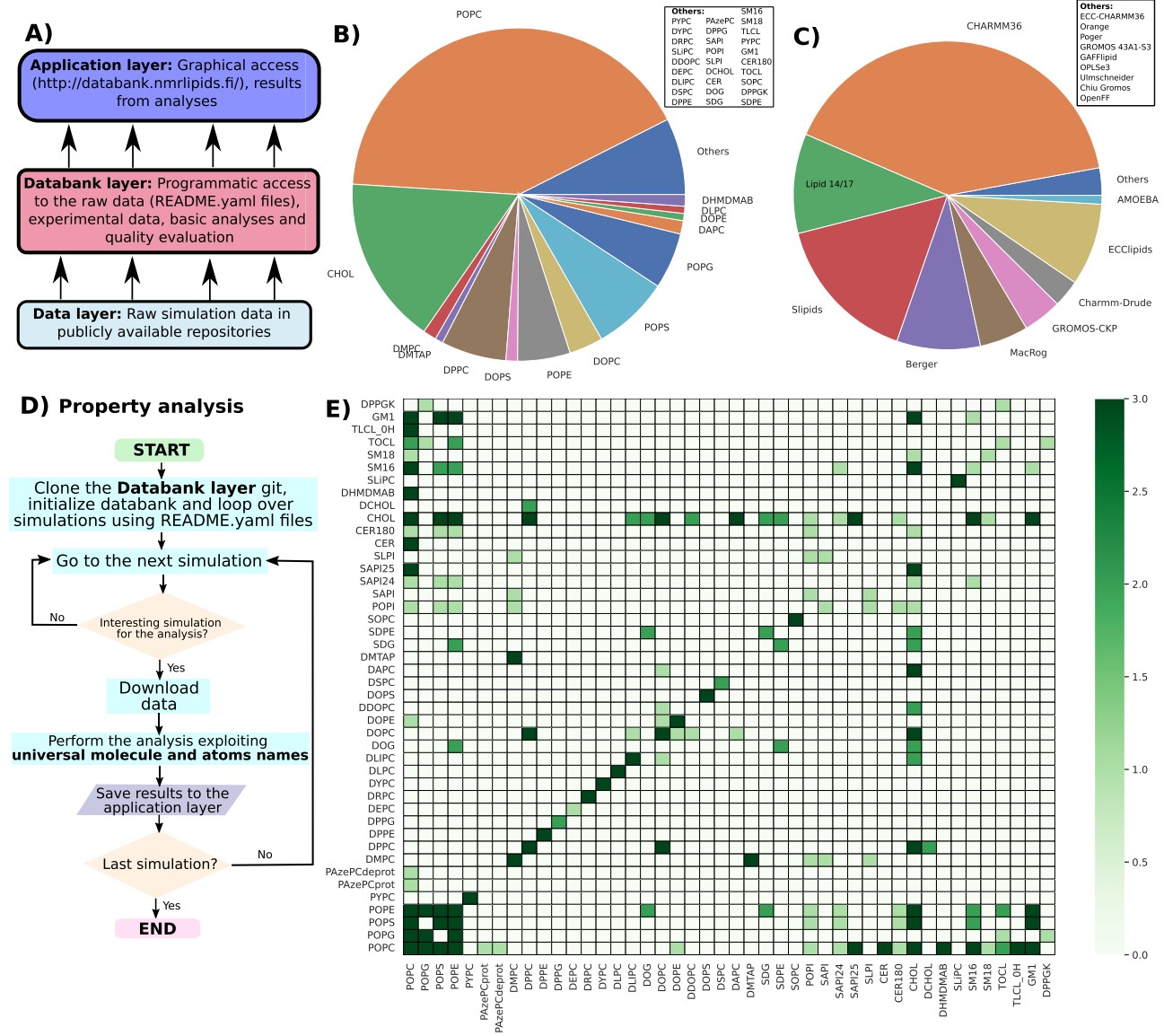

**Fig. 1 | Overview of the NMRlipids Databank. A** Schematic presentation of the overlay structure used in the NMRlipids Databank. A more detailed structure of the *Databank layer* is shown in Supplementary Fig. 1. **B** Distribution of lipids present in the trajectories of the Databank. 'Others' lists lipids occurring in six or fewer simulations. **C** Distribution of force fields in the simulations in the Databank.

References for each force field are given in Supplementary Table 1. **D** Flowchart for performing an analysis of properties through all MD simulations in the NMRlipids Databank using the API. **E** Currently available lipid mixtures in the NMRlipids Databank. Colorbar shows the number of available simulations with the darkest green indicating three or more.

## NMRlipids Databank-API: programmatic access to the MD simulation data

The NMRlipids Databank-API provides programmatic access to all simulation data in the NMRlipids Databank through application programming interface (API). This enables wide range of novel data-driven applications—from construction of ML models that predict membrane properties, to automatic analysis of virtually any property across all simulations in the Databank. The flowchart in Fig. 1D illustrates the practical implementation of such an analysis. After cloning the Databank repository to a local computer, raw data of each simulation can be accessed and analyzed with the help of functions delivered by the NMRlipids Databank-API. Analyses over simulations with different naming conventions can be automatically performed with the help of mapping files that associate the specific naming conventions in each simulation with the universal molecule and atom names used by the Databank. Finally, the analysis results can be stored using the same structure as in the *Databank layer*.

Documentation of the NMRlipids Databank-API and template for new user-defined analyses with further instructions are available at the NMRlipids Databank documentation[19].

While analysis codes and results for basic membrane properties are included in the *Databank layer*, unlimited further analyses can be implemented by anyone in separate repositories in the *Application layer*. When *Application layer* repositories are organised by mimicking the *Databank layer* structure, they can be accessed programmatically and further analyzed using the tools in the NMRlipids Databank-API by implementing the flowchart demonstrated in Supplementary Fig. 2. Novel analyses that demonstrate the power of NMRlipids Databank in selecting the best simulation models, analysing rare phenomena, and extending MD simulations to new fields are implemented in an *Application-layer* repository located at github.com/NMRlipids/DataBankManuscript. The related codes are listed in Supplementary Table 2.

## Selecting simulation parameters using NMRlipids Databank: Best models for most abundant neutral membrane lipids

MD simulations have been particularly useful in understanding membrane systems, although their accuracy has often been compromised by artefacts such as the quality of model parameters[20,21]. Presently, the accuracy of models is becoming increasingly important as researches are progressing from simulations of individual molecules to simulating whole organelles or even cells using interdisciplinary approaches[21–23]. Such systems exhibit intricate emergent behaviour making inaccuracies more difficult to detect, and accumulation of even modest errors may have a dramatic impact on the conclusions drawn. To minimise the detrimental consequences of artificial MD simulation results for their applications, the quality of lipid bilayer MD simulations has to be carefully assessed[20]. This can be done, for example, against the C−H bond order parameters from NMR spectroscopy[16,24] and the form factors from X-ray scattering[13], although it requires comparisons between large number of simulations, which is laborious even with collaborative approaches[6,14–16].

Here we streamlined this process by defining quantitative quality measures for conformational ensembles of individual lipid molecules and membrane dimensions using C−H bond order parameters from NMR and X-ray scattering form factors[13]. These measures enable automatic ranking of lipid bilayer simulations based on their quality against experiments. Qualities of order parameters were evaluated by first calculating the probabilities for each C−H bond order parameter to locate within experimental error, and then averaging the possibilities over different lipid segments ($P^{hg}$, $P^{sn1}$, $P^{sn2}$, and $P^{total}$). Qualities against X-ray scattering experiments ($FF_q$) were estimated as the difference in the experimental and simulated locations of the first form factor minimum. These measures are good proxies for membrane properties because they correlate with the membrane lateral packing and thickness (Fig. 2G, Supplementary Figs. 3 and 4). Ergodicity of conformational sampling of lipids was estimated by calculating $\tau_{rel}$, the convergence time of the slowest principal component divided by the simulation length.

Figure 2 demonstrates how the automatic simulation-quality evaluation and the NMRlipids Databank-API enable rapid selection of the best models for membrane simulations. Figure 2A illustrates that predictions for the lateral packing of membranes composed of two most biologically-abundant neutral membrane lipids, POPC and POPE[7], diverge between different force fields. To find the most realistic parameters to simulate membranes with these lipids, we first ranked all simulations based on order parameter quality (Supplementary Fig. 5), then picked force fields that occur in Fig. 2A (that is: force fields for both POPC and POPE in the Databank), and then ranked them according to the quality of the *sn*-1 chain of POPC (Fig. 2B) and of POPE (Fig. 2C). Simulations with $\tau_{rel}$ clearly above one (larger than 1.3) were discarded in this analysis. Because the average *sn*-1 chain order parameter is a good proxy for the membrane packing (Fig. 2G), rankings in Fig. 2B and C can be used to select the simulations giving the most realistic results in Fig. 2A. Based on this, Lipid17 and Slipids simulations are most realistic for a POPC membrane, while CHARMM36 and GROMOS-CKP simulations predict overly packed bilayers (overestimated order in Supplementary Fig. 6). For POPE, on the other hand, GROMOS-CKP and Slipids are most realistic, while CHARMM36 and Lipid17 predict too packed membranes. In conclusion, the quality evaluation based on the NMRlipids Databank suggests that the Slipids parameters are the best currently available choice for simulations with PC and PE lipids, at least for applications where membrane packing is relevant. Also direct comparisons with the experimental data for the most relevant simulations are shown in Fig. 2D−F and Supplementary Fig. 6A. Figure 2D shows the overall highest-ranked simulation, POPC bilayer with OPLS3e parameters, for the reference.

## Using NMRlipids Databank as a training set for machine learning applications: Predicting multi-component membrane properties

To demonstrate the usage of the NMRlipids Databank to construct ML models that predict membrane properties, we trained a model that predicts area per lipids and thicknesses of membranes with diverse compositions. First we used randomly selected 80% of the Databank simulations to choose the hyperparameters for, and optimise, a set of ML models with the goal of predicting the area per lipid from the membrane composition.

After the parameter optimisation, we tested predictions from different ML models for the area per lipid both against the remaining 20% of the Databank and against areas per lipid reported from simulations of membranes containing mixtures of POPC, POPE, POPS, PI, sphingomyelin lipids, and cholesterol[25–27] that are not included in the Databank. Essential differences between models were not observed when predicting area per lipids of 20% of simulations selected as the test set, but linear regression and Ridge models gave the best correlations with the literature data (predictions from the linear regression model are shown in Fig. 3 and from other models in Supplementary Fig. 7). The linear regression model was selected for further studies due to its simplicity.

To demonstrate the usefulness of the constructed models for understanding multi-component membrane properties, we predicted later packing (areas per lipid) and membrane thicknesses of common biological membranes based on their lipid compositions reported in the literature, see Table 1. The model predicts substantial 50% difference in area per lipid between most densely (influenza virus) and loosely (mitochondria) packed membranes. Difference of 0.8 nm in thickness is predicted between the thinnest (bacterial) and thickest (plasma) membranes. Such differences are expected to effect on many biologically relevant functions of membranes, such as permeation, cholesterol flip-flops (see next sections), and interactions with proteins[28], demonstrating that NMRlipids Databank can be used to give valuable insights on biologically relevant properties of complex biological membranes. Most importantly, the delivered programmatic access to increasing amount of MD simulation data enables training of ML models that predict various membrane properties for all.

## Detecting rare phenomena using NMRlipids databank: cholesterol flip-flops

Lipid flip-flops from one bilayer leaflet to another play an important role in lipid trafficking and regulating membrane properties[7]. Phospholipid flip-flop events are rare when not facilitated by proteins, occurring spontaneously on the timescale of hours or days, while cholesterol, diacylglycerol, and ceramide flip-flop much more often. Still, the reported timescales range from minutes to sub-millisecods[7,29–31]. These timescales were previously accessible only by coarse-grained simulations or free energy calculations[30], and atomistic simulations reporting cholesterol flip-flop events have been published only recently[31–33]. The atomistic studies report an increase in cholesterol flip-flop rates with increasing acyl chain unsaturation level and decreasing cholesterol concentration[31,32], but the amount of data in these individual studies was not sufficient to systematically assess correlations between cholesterol flip-flop rates and membrane properties. Here, we demonstrate that the NMRlipids Databank-API makes analyses of such rare phenomena accessible for all by enabling access to a large amount of MD simulation data as illustrated in Fig. 1. This is particularly useful for scientists in various fields of science and industry who lack access to the computational resources or the expertise to produce the large amounts of MD simulation data required for such analyses.

Using the general workflow depicted in Fig. 1D, we first calculated the flip-flop rates from all the simulations available in the NMRlipids Databank. Flip-flops were observed for cholesterol, DCHOL

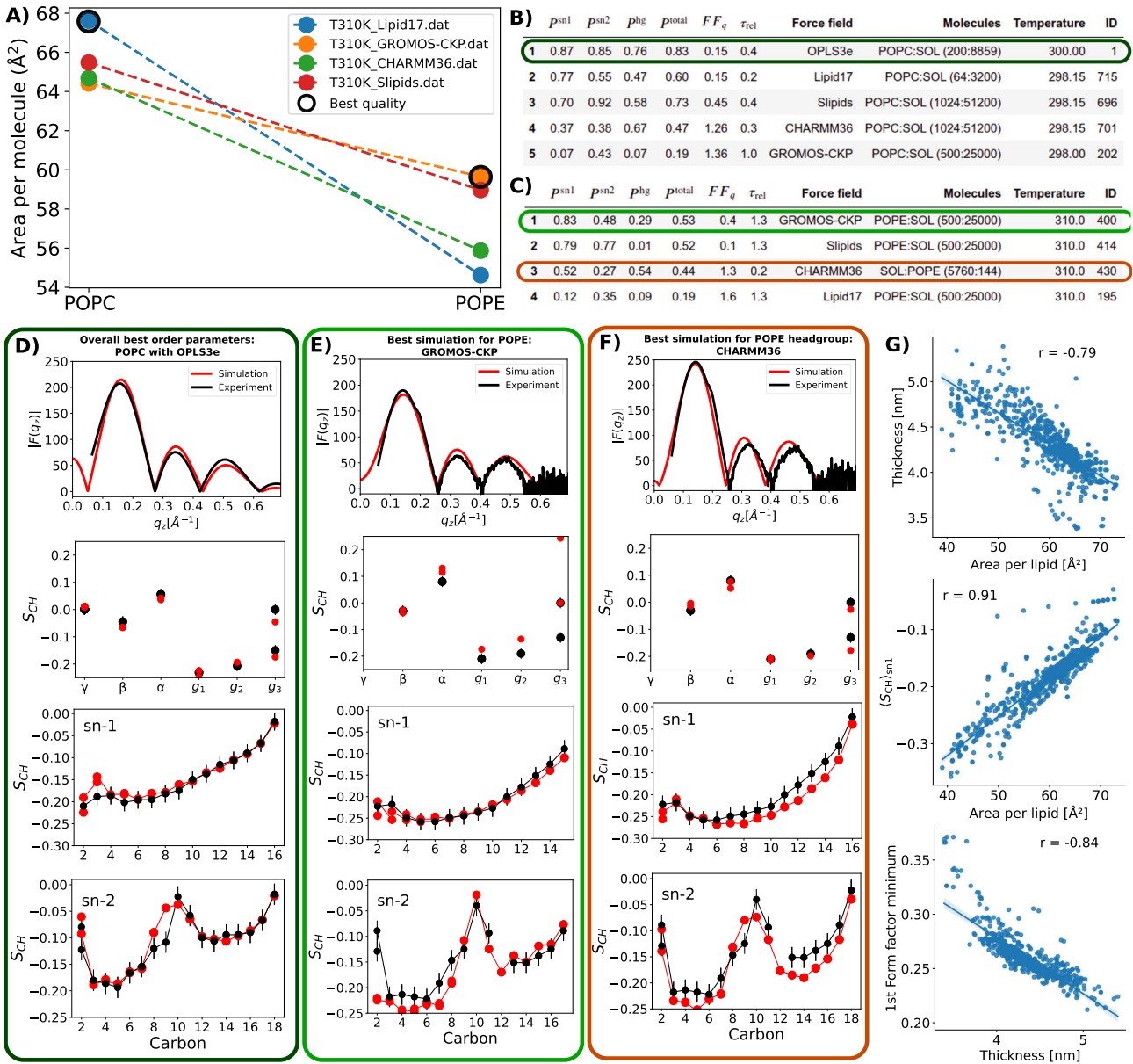

**Fig. 2 | Examples of data obtained using NMRlipids Databank. A** Area per lipid of POPC and POPE lipid bilayers predicted by different force fields at 310 K in simulations that are available in the NMRlipids Databank. The data points from the best-performing simulations, based on rankings in (**B**, **C**), are surrounded by black circles. **B** Best POPC simulations ranked based on the *sn*-1 acyl chain order parameter quality ($P^{sn1}$). Also *sn*-2 acyl chain ($P^{sn2}$), headgroup ($P^{hg}$) and total ($P^{total}$) order parameter qualities, form factor quality ($FF_q$), and relative equilibration time for conformations ($\tau_{rel}$) are shown. Note that the best possible order parameter quality is one, while the best possible form factor quality is zero. **C** Best POPE simulations ranked based on the *sn*-1 acyl chain order parameter quality. Direct comparison against experimental (NMR order parameters and X-ray scattering) data exemplified for a simulation with the best overall order parameter quality (**D**), the best quality for POPE lipid (**E**), and the headgroup quality for POPE (**F**). Error bars for simulations are standard error of the mean over different lipids (*n* = number of lipids in a simulation shown in **B**). Error bars for experiments are 0.02[13]. **G** Scatter plots and Pearson correlation coefficients, *r*, for the membrane area per lipid, thickness, first minimum of X-ray scattering form factor and average order parameter of the *sn*-1 acyl chain extracted from the NMRlipids Databank. All correlation coefficients have *p*-value below 0.001 with two-sided test. For more correlations see Supplementary Fig. 3.

(18,19-di-nor-cholesterol), DOG (1,2-dioleoyl-*sn*-glycerol), and SDG (1-stearoyl-2-docosahexaenoyl-*sn*-glycerol). The observed cholesterol flip-flop rates, ranging between 0.001–1.6 µs⁻¹ with the mean of 0.16 µs⁻¹ and median of 0.07 µs⁻¹, are in line with the previously reported values from atomistic MD simulations[31–33]. The flip-flop rate of DCHOL, 0.2 µs⁻¹, was close to the average value of cholesterol, while the average rates for diacylglycerols DOG (0.4 µs⁻¹) and SDG (0.5 µs⁻¹) were higher than for cholesterol. Flip-flops were not observed for other lipids, giving the upper limits for PC-lipid flip-flop rate as 9 × 10⁻⁶ µs⁻¹ and for ceramide (*N*-palmitoyl-D-*erythro*-sphingosine) as 0.002 µs⁻¹. Thus, the available data in the NMRlipids Databank suggest

that the lipid flip-flop rate decreases in the order: diacylglycerols > cholesterol > other lipids including ceramides. However, the amount of data for diacylglycerols (8 simulations with the Lipid17 force field) and ceramide (3 simulations with CHARMM36) is less than that for cholesterol (83 simulations); thus we cannot fully exclude the effect of force field or composition on this comparison.

Nevertheless, we used the general workflow depicted in Supplementary Fig. 2 to analyse how the flip-flop rates calculated from the NMRlipids Databank depend on membrane properties. Figure 4B−D show cholesterol flip-flop rates and their histograms as a function of membrane thickness, lateral density, and acyl chain order. The results

reveal a non-linear correlation between cholesterol flip-flop rate and membrane packing (depicted as area per lipid): Flip-flop rates increase by an order of magnitude when membrane packing density decreases, and a major jump is observed at low membrane packing. Such order-of-magnitude changes in cholesterol flip-flop rate with the membrane composition may have major implications in understanding lipid trafficking and membrane biochemistry[31,33]. Because the results from the NMRlipids Databank are averaged over a large range of membrane compositions and force fields, they show that the strong dependence of cholesterol flip-flop rate on membrane properties is not limited to the particular lipid compositions or force fields used in the previous studies[31-33].

### Extending the scope of MD simulations to new fields using NMRlipids Databank: Water diffusion anisotropy in membrane systems

The anisotropic diffusion of water and hydrophilic molecules in directions parallel and perpendicular to membranes is an important parameter in models describing the translocation of drugs through biological material, particularly in the skin[12,34-36]. Water anisotropic diffusion plays a role also in the signal formation in diffusion-tensor MRI imaging[11]. MD simulations are rarely used to analyze the anisotropic diffusion of water, since only a few membrane permeation events of water are typically observed in a single MD simulation trajectory[37,38], thereby making the collection of a sufficient amount of data challenging. Here, we show that the API access to the data in NMRlipids Databank enables systematic analysis on how the anisotropic diffusion of water depends on membrane properties in multilamellar membrane systems, thereby extending the application of MD simulations to new fields.

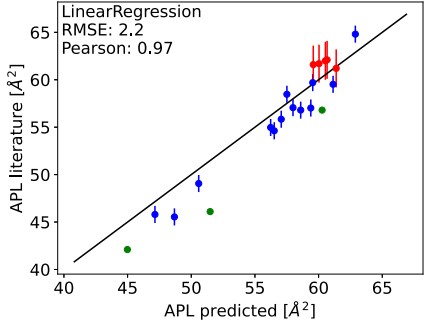

**Fig. 3 | Predictions of areas per lipid (APL) of multi-component membranes composed of POPC, POPE, POPS, PI, sphingomyelin lipids, and cholesterol from linear regression model against literature data from simulations (green[25], blue[26], and red[27]).** Error bars are from the same publications as the values. Black line indicates $x = y$.

To this end, we first calculated the water permeability through membranes from all simulations in the NMRlipids Databank using the general workflow depicted in Fig. 1D. The resulting non-zero values range between 0.3 and 322 μm/s with the mean of 14 μm/s and median of 8 μm/s. These values agree with the previously reported simulation results[37,38], but are on average larger than experimental values reported for PC lipids in the liquid crystalline phase, 0.19–0.33 μm/s[39]. Using the workflow depicted in Supplementary Fig. 2, we then plotted the observed permeabilities and their histogrammed values in Fig. 5B–E as a function of temperature, membrane thickness, area per lipid, and acyl chain order. As expected, the permeability increases with the temperature, giving an average energy barrier of $14 \pm 3\ k_BT$ for the water permeation from the Arrhenius plot in Fig. 5B. On the other hand, the water permeability on average decreases when membranes become more packed, that is, with decreasing area per lipid and increasing thickness and acyl chain order (Fig. 5C–E). Permeation of water through bilayers depends on membrane properties also according to previous studies, but there is no established consensus on whether the area per lipid[40] or bilayer thickness[41] is the main parameter determining the permeability. Our analysis over the NMRlipids Databank, containing significantly more data than what was available in previous studies, suggest non-linear dependencies on both of these parameters. Clear dependencies of permeability on hydration level or the fraction of charged lipids, cholesterol, or POPE in the membrane were not observed (Supplementary Fig. 8).

To examine how water diffusion anisotropy depends on membrane properties in a multi-lamellar lipid bilayer system, we analyzed the water diffusion parallel to the membrane surface from all simulations in the NMRlipids Databank using the general workflows depicted in Fig. 1D and Supplementary Fig. 2. The parallel diffusion coefficient of water, $D_\parallel$, decreases with reduced hydration and increases with the temperature, but dependencies on the membrane area per lipid, thickness, or fraction of charged lipids were not observed in Fig. 5 and Supplementary Fig. 9. Simulation results are close to the experimental values with low hydration levels in Fig. 5F, but increase to approximately 50% higher than the experimental value for bulk water diffusion value ($3.1 \times 10^{-9}$ m²/s at 313 K[42]) with high hydration levels. This is not surprising as the most common water model used in membrane simulations, TIP3P, overestimates the bulk water diffusion[43]. To estimate the diffusion anisotropy of water, $D_\perp/D_\parallel$, in multilamellar membrane system, the permeability coefficients of water through membranes were translated to perpendicular diffusion coefficients, $D_\perp$, using the Tanner equation[44,45]. The resulting perpendicular diffusion coefficients are approximately five orders of magnitude smaller than the lateral diffusion coefficients of water (Fig. 5G, H), which is at the upper limit of anisotropy estimated from experimental data[12]. A significant increase in the diffusion anisotropy with membrane packing is observed, as $D_\perp/D_\parallel$ deviates further from unity with decreasing area per lipid and increasing thickness in Fig. 5G, H. This follows from

**Table 1 | Areas per lipid (APL) and membrane thicknesses predicted by the linear regression model trained using the NMRlipids Databank for membrane compositions corresponding different biological membranes**

| Membrane | Composition [lipid(%-fraction)] | APL[Å²] | Thickness [nm²] |
|---|---|---|---|
| Mitochondria | POPC(37) : POPE(31) : PI(6) : CL(22) : CHOL(4)[7,70,71] | 74 (66)[a] | 4.4 |
| Bacterial | POPC(20) : POPE(35) : POPG(35) : CL(5) : CHOL(5)[72] | 61 (60)[a] | 4.2 |
| ER | POPC(54) : POPE(20) : POPS(4) : PI(11) : SM(4) : CHOL(8)[7,70,71] | 57 | 4.6 |
| Golgi | POPC(36) : POPE(21) : POPS(6) : PI(12) : SM(7) : CHOL(18)[7,70,71] | 52 | 4.8 |
| Plasma | POPC(23) : POPE(11) : POPS(8) : PI(7) : SM(17) : CHOL(34)[7,70,71] | 45 | 5.0 |
| Synaptic | POPC(28) : POPE(20) : POPS(9) : PI(2) : SM(4) : CHOL(37)[73] | 45 | 4.8 |
| Influenza | POPC(5) : POPE(32) : POPS(15) : SM(9) : CHOL(40)[74,75] | 42 | 4.8 |

Systems are sorted in the order of increasing membrane packing. Compositions of different membranes are estimated based on references given in the composition column.
[a]Value in parenthesis is the area per acyl chain taking into account that cardiolipin has four chains per molecule while other lipids have two.

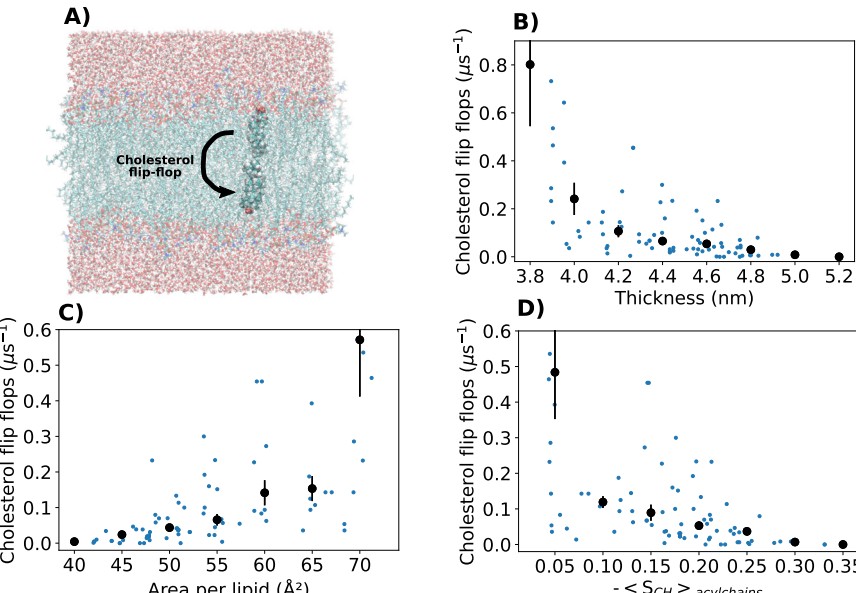

**Fig. 4 | Quantification of cholesterol flip-flop events in NMRlipids Databank simulations. A** Illustration of cholesterol flip-flop. **B**–**D** Cholesterol flip-flops analyzed from the Databank as a function of membrane thickness, area per lipid, and acyl chain order. Values from simulations with non-zero flip-flop rates are shown with blue dots. Histogrammed values are shown with black dots. For the mean value in each bin, average weighted with the simulation lengths was used, and error bars show the standard error of the mean.

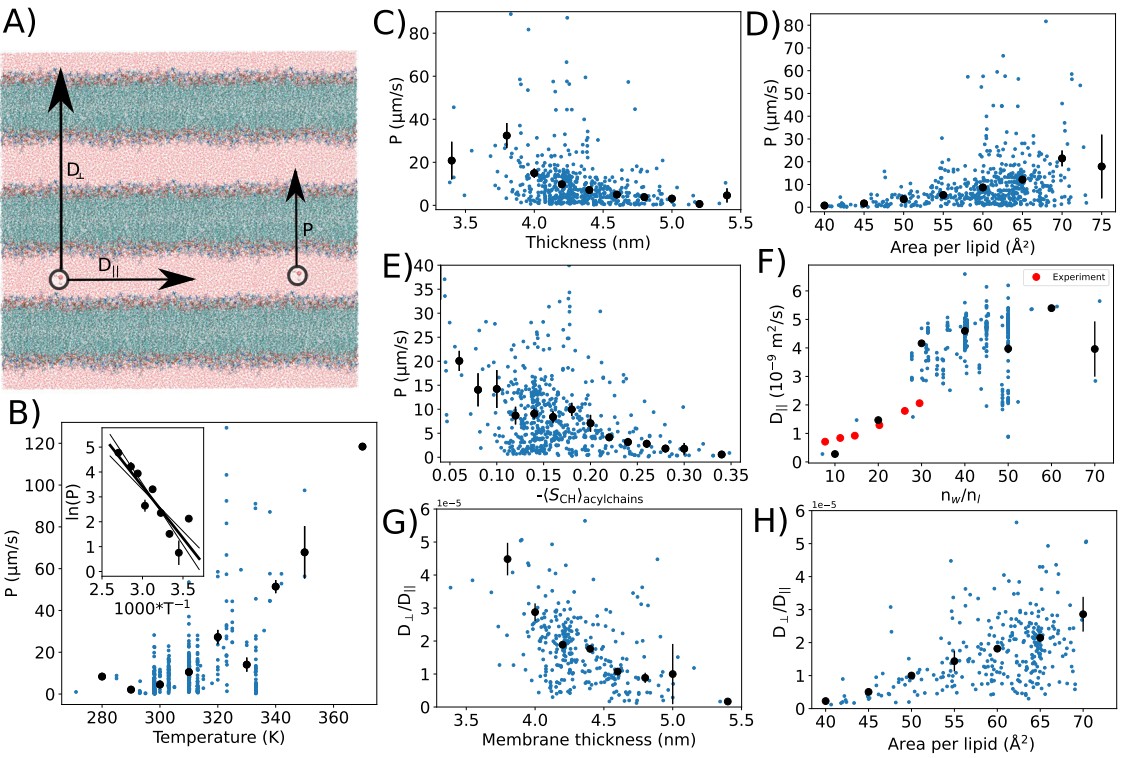

**Fig. 5 | Quantification of water diffusion in NMRlipids Databank simulations. A** Water diffusion, $D_\perp$, and permeability, $P$, through membranes, and lateral diffusion along the membrane, $D_\parallel$, illustrated in a multilamellar stack of lipid bilayers. **B**–**E** Water permeation through membranes analyzed from the Databank as a function of temperature, thickness, area per lipid, and acyl chain order. Inset in (**B**) shows the Arrhenius plot of permeation ($\ln(P)$ vs. $1/T$) that gives $14 \pm 3\ k_BT$ for the average activation energy for water permeation through lipid bilayer. **F** Lateral diffusion of water as a function of hydration level. Experimental points for DMPC bilayers at 313 K at different hydration levels are shown[76]. **G**, **H** Diffusion anisotropy of water as a function of thickness and area per lipid. Non-zero permeation and diffusion values from simulations are shown with blue dots. Histogrammed values are shown with black dots. For the mean value in each bin, average weighted with the simulation lengths was used, and error bars show the standard error of the mean. Only bins with more than one microsecond of data in total were used for water permeation.

decreasing water permeability with membrane packing (Fig. 5C, D), while lateral diffusion remains approximately constant (Supplementary Fig. 9A, C).

In summary, our results suggest that the bilayer packing has a substantial effect on anisotropic water diffusion in multi-membrane lipid systems. The several-fold larger anisotropy in membranes with higher lateral density is expected to play a role in pharmacokinetic models not only for water but also for other hydrophilic molecules[12]. Furthermore, the enhanced understanding of this anisotropy may help in developing new diffusion-tensor-based MRI imaging methods where signals originate from the anisotropic diffusion of water in biological matter[11].

## Discussion

Sharing of biomolecular MD simulation and other data is becoming increasingly important in the age of big data and AI[1,4]. Besides the data itself, also programmatic access is a necessary requirement for data-driven and ML applications. This is particularly challenging when field-specific smart guidelines and community-consensus best-practices have not yet been defined, which is the case in biomolecular simulations[4]. The NMRlipids Databank demonstrates how these issues can be solved by the overlay databank design, where the raw data are distributed to already publicly available decentralised locations, while the core of the databank is composed only of the metadata stored in a version-controlled git repository with an open-access license. On the other hand, the open-collaboration approach developed in the NMRlipids Project[6] creates incentives for sharing the data by offering authorship in published articles to the contributors. Advantages of such an approach are demonstrated here for membrane simulations, yet the concept can be applied to any other biomolecules, as well as in any other field where similar barriers hinder the AI revolution, such as the assignment of NMR spectra[5].

The NMRlipids Databank-API delivers programmatic access to MD simulation data that can be used as training set for diverse data-driven and ML applications that predict membrane properties. Such applications could be analogous to AlphaFold[3] and other tools[46,47] that predict protein structures from their sequence using AI. This is demonstrated here by building ML models to predict multi-component membrane properties. Furthermore, the analysis of cholesterol flip-flop events (Fig. 4) and water permeation through membranes (Fig. 5) demonstrate how a large amount of accessible simulation data in terms of quantity (e.g., simulation length and number of conformations) and content (e.g., lipid compositions and ion concentrations) enable analyses of rare phenomena that are beyond the current possibilities for a single research group. Such

analyses also pave the way for applications of MD simulations in new fields, as demonstrated here by analysing an essential parameter in pharmacokinetic modelling and MRI imaging:[11,12] the anisotropic diffusion of water in membrane systems (Fig. 5). These possibilities are particularly valuable for scientists who do not typically have access to large-scale MD simulation data.

The focus of biomolecular simulations is moving from studies of individual molecules to larger complexes and even whole cells and organelles[21–23]. Simultaneously, machine-learning-based models for predicting the behaviour of biomolecules and automatic approaches to parametrise models are emerging[3,20]. The resources delivered by the NMRlipids Databank will support developments in both of these directions. Automatic quality evaluation and ranking of simulations against experimental data enable the selection of best simulations for specific applications without laborious manual force field evaluation. This also streamlines automatic parametrization procedures for atomistic and coarse grained simulations by, for example, pinpointing typical failures of force fields and highlighting points of improvement. Such practises for fostering the accuracy of simulations are becoming increasingly important as small errors accumulate when complexity and size of simulated systems are increasing. Examples of impact of NMRlipids and other overlay databanks in different disciplines are listed in Table 2, yet the scope of applications is expected to further widen with increasing amount of publicly shared data.

## Methods
### Structure of the databank

The overlay structure designed for the NMRlipids Databank is composed of three layers (Fig. 1A). The *Data layer* contains raw data that can be distributed to publicly available servers such as Zenodo (zenodo.org). The core content of the Databank locates in the *Databank layer*, which is a git repository at github.com/NMRlipids/Databank and is also permanently stored in a Zenodo repository (https://doi.org/10.5281/zenodo.7875567). The essential information of each simulation is stored in a human-and-machine-readable README.yaml file located in a subfolder of the /Data/Simulations folder in the *Databank layer* repository; each subfolder has a unique name constructed based on a hash code of the trajectory and topology files of each simulation. The README.yaml files in these folders contain access to all information that is needed for further analysis of simulations, such as links to the raw data and associations with the universal molecule and atom names. The content of these files is described in detail in Supplementary Table 3 and in the NMRlipids Databank documentation nmrlipids.github.io. Results from analyses of basic membrane properties (area per lipid, thickness, C−H bond order

**Table 2 | Examples of impact of NMRlipids and other overlay databanks in different disciplines**

| Target group | Outcomes | Practical examples |
|---|---|---|
| Computational and data scientists | Databanks with programmatic access, training sets for machine learning applications, automatic optimisation of simulation models | Programmatic access to all publicly available MD simulation trajectories, machine learning models predicting properties of complex biomolecular assemblies (Table 1), optimising force field and model parameters from atomistic to continuum scale |
| Biomedical scientists | Applications of biomolecular modelling in new fields, coupling omics to biomolecular structure and dynamics | Anisotropic water diffusion for pharmacokinetics and MRI imaging applications (Fig. 5), properties of complex cellular structures based on composition analyses from omics |
| Material scientists | Predictions of complex biomolecular material properties from data-driven and machine learning models | Optimising composition of lipid nanoparticle formulations for desired properties, predictions of bioinspired material properties |
| Biophysicists | Novel analyses and predictions for properties of biomolecular assemblies and their compositions | Analyses of rare phenomena (for example, Lipid flip-flops and water permeation in Figs. 4 and 5), data-driven analyzes and machine learning models predicting correlations between properties and compositions of biomolecular assemblies (Figs. 2 and 5) |
| Students and teachers | Graphical and programmatic access to biomolecular structure and dynamics | Illustration of disorder and dynamics in biomolecules, example data for bioinformatics and data-science courses |
| Reviewers and publishers | Tool to facilitate FAIRness of data, transparency and reproducibility | All publicly available data can be included in an overlay databank irrespectively of its location |

parameters, X-ray scattering form factors, and relaxation of principal components) are stored in the same folders as the README.yaml files. Experimental data used for ranking is stored in the `/Data/experiments` folder, the ranking results in `/Data/Ranking`, and the relevant scripts in the `/Scripts/` folder in the *Databank layer* repository. The scripts in the NMRlipids Databank are mainly written in Python and many of them use the MDAnalysis module[48,49]. The Databank structure is illustrated in more detail in Supplementary Figure 1. Whenever specific files or folders are referred here, they locate at the *Databank layer* repository unless stated otherwise.

### Universal naming convention for molecules and atoms

When analysing simulation trajectories, atoms and molecules often need to be called by the names used in the trajectory. However, these names typically vary between force fields, as a universal naming convention has not been defined for lipids. To enable automatic analyses over all the simulations in the NMRlipids Databank, we have defined universal naming conventions for the molecules and atoms therein. The universal abbreviations used in the NMRlipids Databank for each molecule are listed in the NMRlipids Databank documentation[18]. The atom names used in simulation trajectories are connected to the universal atom names using the mapping files defined in the NMRlipids Project[18]. These files are located at `/Scripts/BuildDatabank/mapping_files` in the NMRlipids Databank repository. These files also define whether an atom belongs to the headgroup, glycerol backbone, or acyl chain region in a lipid. In practise, the force-field-specific molecule names and mapping file names are defined in the README.yaml files for each molecule in each simulation as described in the NMRlipids Databank documentation[50].

### Adding data to NMRlipids Databank

The NMRlipids Databank is open for additions of simulation data by anyone. In practise, the required information is first manually entered into an info.yaml file that is then added into the `/Scripts/BuildDatabank/info_files` folder trough a git pull request. Rest of the information to be stored in the README.yaml files will be then automatically extracted using the `/Scripts/BuildDatabank/AddData.py` script. The required manually entered and automatically extracted information are described in detail in Supplementary Table 3 and in the NMRlipids Databank documentation[50]. The documentation includes also detailed instructions to add data. To avoid ineligible entries and minimise human errors, the pull requests are monitored before the acceptance and generation of the README.yaml files. Currently, the NMRlipids Databank is composed of simulations that are found from the Zenodo repository with an appropriate license; most, but not all, of these trajectories originate from previous NMRlipids projects[6,14–16].

### Experimental data

Experimental data used in the quality evaluation, currently composed of C–H bond order parameters and X-ray scattering form factors, are stored in `/Data/experiments` in the NMRlipids Databank repository. Similarly to simulations, each experimental data set has a README.yaml file containing all the relevant information about the experiment. The keys and their descriptions for the experimental data, as well as detailed instructions to add the date, are given in Supplementary Table 4 and in the NMRlipids Databank documentation[51]. The NMR data currently in the NMRlipids Databank are taken from refs. 15,16,52–56 and the X-ray scattering data from refs. 56–62. In addition, previously unpublished NMR data for POPE, POPG, and DOPC was acquired as described in Supplementary Figs. 10–15 and contributed to the Databank.

### Analysing simulations

In practise, simulations in the NMRlipids Databank can be analyzed by executing a programme that (1) loops over the README.yaml files in the *Databank layer*, (2) downloads the data using the information in the README.yaml files, and then (3) performs the desired analysis on a local computer utilising the universal naming conventions for molecules and atoms defined in the README.yaml and mapping files. This general procedure is illustrated in Fig. 1D, and a templates for user-defined analyses are available via NMRlipids Databank documentation[19]. Further practical examples of codes performing such analyses are listed in Supplementary Tables 2 and 5. The equilibration period given by the user (TIMELEFTOUT in Supplementary Table 3) is discarded from the trajectories in all analysis codes. For further details, see the NMRlipids Databank documentation nmrlipids.github.io.

### Principal components analysis of equilibration of simulations

To estimate how well conformational ensembles of lipids are converged in trajectories, the Principal Component Analysis (PCA) following the PCALipids protocol was used[63,64]. To this end, each lipid configuration was first aligned to the average structure of that lipid type, and PCA analysis was then applied on the Cartesian coordinates of all heavy atoms of the lipid. Because the motions along the first, major, principal component are the slowest ones[63], the equilibration of each lipid type was estimated from the ratio between the distribution convergence of the trajectories projected on the first PC and the trajectory length, $\tau_{rel} = \tau_{convergence}/\tau_{sim}$[63,64]. If $\tau_{rel} < 1$, simulations can be considered to be sufficiently long for the lipid molecules to have sampled their conformational ensembles, while in simulations with $\tau_{rel} > 1$ individual molecules may not have fully sampled their conformational ensembles. Rigid molecules that do not exhibit significant conformational fluctuations, such as sterols, were excluded from the analysis. In practise, the distribution convergence times were calculated utilising its linear dependence on autocorrelation decay times, $\tau_{convergence} = k\tau_{autocorrelation}$, because calculation of autocorrelation decay times is faster and computationally more stable than direct calculation of distribution convergence times[63,64]. The empirical coefficient $k = 49$ was calculated based on the analysis of 8 trajectories with the length of more than 200 ns, including simulations of POPC, POPS, POPE, POPG, and DPPC with the CHARMM36 force field. Because the coefficient $k$ does not depend on the force field[63], the value determined from these CHARMM36 simulations can be used for all simulations in the Databank. The script that calculates the equilibration of lipids is available at `Scripts/BuildDatabank/NMRPCA_timerelax.py` in the NMRlipids Databank repository. The resulting values are stored in files named `eq_times.json` at folders in `/Data/Simulations` in the NMRlipids Databank repository.

### Calculation of C–H bond order parameters

The C–H bond order parameters were calculated directly from the carbon and hydrogen positions using the definition

$$S_{CH} = \frac{1}{2}\langle 3\cos^2\theta - 1 \rangle, \qquad (1)$$

where angular brackets denote the ensemble average, i.e., average over all sampled configurations of all lipids in a simulation, and $\theta$ is the angle between the C–H bond and the membrane normal. As in previous NMRlipids publications, the order parameters were first calculated separately for each lipid and the standard error of the mean over different lipids was used as the error estimate[6]. However, order parameters for simulations with $\tau_{rel} > 1$ may be influenced by the starting structure and thereby their error bars may be underestimated. The script that calculates C–H bond order parameters from all simulations in the NMRlipids Databank is available at `/Scripts/AnalyzeDatabank/calcOrderParameters.py` in the NMRlipids Databank repository. The resulting order parameters are stored for all simulations in files named `[lipid_name]`

OrderParameters.json at folders in /Data/Simulations in the NMRlipids Databank repository.

## Calculation of X-ray scattering form factors

X-ray scattering form factors were calculated using the standard equation for lipid bilayers that does not assume symmetric membranes[13],

$$F(q) = \left| \int_{-D/2}^{D/2} \Delta\rho_e(z) \exp(izq_z) dz \right|, \quad (2)$$

where $\Delta\rho_e(z)$ is the difference between the total and solvent electron densities, and $D$ is the simulation box size in the $z$-direction (normal to the membrane). For the calculation of density profiles, atom coordinates were first centred around the centre of mass of lipid molecules for every time frame, and a histogram of these centred positions, weighted with the number of electrons in each atom, was then calculated with the bin width of 1/3 Å. Electron density profiles were then calculated as an average of these histograms over the time frames in simulations. The script to calculate form factors for all simulations in the NMRlipids Databank is available at Scripts/AnalyzeDatabank/calc_FormFactors.py. The resulting form factors are stored for all simulations in files named FormFactor.json at folders in /Data/Simulations in the NMRlipids Databank repository.

## Calculation of area per lipid and bilayer thickness

Area per lipids of bilayers were calculated by dividing the time-averaged area of the simulation box with the total number of lipids and surfactant molecules in the simulation. The script that calculates the area per lipid from all simulations in the NMRlipids Databank repository is available at Scripts/AnalyzeDatabank/calcAPL.py in the NMRlipids Databank repository. The resulting area per lipids are stored for all simulations in files named apl.json at folders in /Data/Simulations.

Thicknesses of lipid bilayers were calculated from the intersection points of lipid and water electron densities. The script that calculates the thicknesses of all simulations in the NMRlipids Databank is available at Scripts/AnalyzeDatabank/calc_thickness.py in the NMRlipids Databank repository. The resulting thicknesses are stored in files named thickness.json at folders in /Data/Simulations in the NMRlipids Databank repository.

## Quality evaluation of C–H bond order parameters

As the first step to evaluate simulation qualities against experimental data, a simulation is connected to an experimental data set if the molar concentrations of all molecules are within ± 3 percentage units, charged lipids have the same counterions, and temperatures are within ± 2 K. For molar concentrations of water, the exact hydration level is considered only for systems with molar water-to-lipid ratio below 25, otherwise the systems are considered as fully hydrated. In practise, the connection is implemented by adding the experimental data path into the simulation README.yaml file using the /Scripts/BuildDatabank/searchDATABANK.py script in the NMRlipids Databank repository.

The quality of each C–H bond order parameter is estimated by calculating the probability for a simulated value to locate within the error bars of the experimental value. Because conformational ensembles of individual lipids are assumed to be independent in a fluid lipid bilayer, $\frac{S_{CH}-\mu}{s/\sqrt{n}}$ has a Student's t-distribution with $n-1$ degrees of freedom and $\mu$ representing the real mean of the order parameter. The probability for an order parameter from simulation to locate within

experimental error bars can be estimated from equation

$$P = f\left(\frac{S_{CH} - (S_{exp} + \Delta S_{exp})}{s/\sqrt{n}}\right) - f\left(\frac{S_{CH} - (S_{exp} - \Delta S_{exp})}{s/\sqrt{n}}\right), \quad (3)$$

where $f(t)$ is the Student's t-distribution, $n$ is the number of independent sample points for each C–H bond (which equals the number of lipids in a simulation), $S_{CH}$ is the sample mean from Eq. (1), $s$ is the variance of $S_{CH}$ calculated over individual lipids, $S_{exp}$ is the experimental value, and $\Delta S_{exp}$ its error. The error of $\Delta S_{exp} = 0.02$ is currently assumed for all experimental order parameters[13], yet more accurate ones may be available in the future[65]. Because a lipid bilayer simulation contains at least dozens of lipids, the Student's t-distribution could be safely approximated with a normal distribution. However, with the quality of currently available force fields, the simulation values can be so far from experiments that a normal distribution leads to probability values below the numerical accuracy of computers. To avoid such numerical instabilities, we opted to use the first order Student's t-distribution that has slightly higher probabilities for values far away from the mean. On the other hand, some force fields exhibit too slow dynamics, which leads to large error bars in the $S_{CH}$ values[66]. Such artificially slow dynamics widens the Student's t-distribution in Eq. (3), thereby increasing the probability to find the simulated value within experimental error bars. Therefore, the $S_{CH}$ with simulation error bars above the experimental error 0.02 are not included in the quality evaluation.

To streamline the comparison between simulations, we define the average qualities for different fragments (frag = 'sn-1', 'sn-2', 'head-group', or 'total', with the last referring to all order parameters within a molecule) within each lipid type in a simulation as

$$P^{frag}[lipid] = \langle P[lipid]\rangle_{frag} F_{frag}[lipid], \quad (4)$$

where $\langle P[lipid]\rangle_{frag}$ is the average of the individual $S_{CH}$ qualities within the fragment, and $F_{frag}[lipid]$ is the percentage of order parameters for which the quality is available within the fragment. The overall quality of different fragments in a simulation (frag = 'tails', 'headgroup', or 'total') are then defined as a molar-fraction-weighted average over different lipid components

$$P^{frag} = \sum_{lipid} \chi_{lipid} P^{frag}[lipid], \quad (5)$$

where $\chi_{lipid}$ is the molar fraction of a lipid in the bilayer and 'tails' refer to the average of all acyl chains.

The quality evaluation of order parameters is implemented in /Scripts/BuildDatabank/QualityEvaluation.py in the NMRlipids Databank repository. The resulting qualities for each $S_{CH}$ are stored in files named [lipid_name]_OrderParameters_quality.json, for individual lipids in files named [lipid_name]_FragmentQuality.json, and for overall quality for fragments in files named system_quality.json at folders in /Data/Simulations in the NMRlipids Databank repository.

## Quality evaluation of X-ray scattering form factors

Because experiments give form factors only on a relative intensity scale, they should be scaled before comparing with the simulation data. Here we use the scaling coefficient for experimental intensities defined in the SIMtoEXP program[67]

$$k_e = \frac{\sum_{i=1}^{N_q} \frac{|F_s(q_i)||F_e(q_i)|}{(\Delta F_e(q_i))^2}}{\sum_{i=1}^{N_q} \frac{|F_e(q_i)|^2}{(\Delta F_e(q_i))^2}}, \quad (6)$$

where $F_s(q)$ and $F_e(q)$ are form factors from a simulation and experiment, respectively, $\Delta F_e(q)$ is the error of the experimental form factor, and summation goes over the experimentally available $N_q$ points.

Also, a quality measure based on differences in the simulated and experimental form factors across the available $q$-range is defined in the SIMtoEXP program[67]. However, the lobe heights in the simulated form factors depend on the simulation box size, as shown in Supplementary Fig. 4; consequently, the quality measure defined in SIMtoEXP would also depend on the simulation box size. In contrast, locations of the form factor minima (or, in precise terms: the minima of the absolute value of the form factor) are independent of the simulation box size (Supplementary Fig. 4). Here we use only the location of the first form factor minimum for quality evaluation, because (due to fluctuations) the location of the second minimum is difficult to detect automatically in some experimental data sets, such as the POPE data in Fig. 2E, F. The first minimum correlates well with the thickness of a membrane (Fig. 2G), although the correlation of the second minima would be even stronger (Supplementary Fig. 3). In practise, we first filter the fluctuations from the form factor data using the Savitzky–Golay filter (window length 30 and polynomial order 1) and locate the first minimum at $q > 0.1\,\text{Å}^{-1}$ from both simulation ($FF^{sim}_{min}$) and experiment ($FF^{exp}_{min}$). The quality of a form factor is then defined as the Euclidean distance between the minima locations: $FF_q = |FF^{sim}_{min} - FF^{exp}_{min}| \times 100$.

The quality evaluation of form factors is implemented in `/Scripts/BuildDatabank/QualityEvaluation.py` in the NMRlipids Databank repository. The resulting form factor qualities are stored in files named `FormFactorQuality.json` at folders in `/Data/Simulations` in the NMRlipids Databank repository.

## Training machine learning model to predict properties of multi-component membranes

Different ML approaches were tested to find the best model to predict the area per lipids of multi-component membranes from their lipid composition. To his end, we tested linear, Lasso, Ridge, elastic net, decision tree, random forest, multi-layer perceptron, k-nearest neighbour, gradient boosting, AdaBoost, and XGBoost regressors. The models were trained to predict the area per lipid for a membrane with the given molar fractions of membrane lipids. Most abundant lipids in the Databank, POPC, POPE, POPG, POPS, and cholesterol, were considered explicitly, while SM, cardiolipins, and PI lipids with more sparse data were all grouped together based on their headgroups independently on acyl chain content. Salt concentration, temperature and other conditions were not considered.

To find the optimal model for this task, we first randomly divided simulations into training (80% of the simulations) and test (20% of the simulations) sets. Then, we used the training set with 5-fold cross validation to conduct a grid search for hyperparameters for the set of ML regression models. The best performing hyperparameters were then used to fit each model to the training set. The performance of the models was then tested on the remaining 20% of the data extracted from the Databank, and additionally, by predicting area per lipids from multi-component membrane simulations reported in the literature but not included in the Databank[25–27]. Due to its simplicity and good performance in these tests, we selected to use linear regression model for further studies. The linear regression model predicting membrane thicknesses was trained similarly. The ML models and analysis were implemented with Python using scikit-learn which was supplemented with the xgboost library to include extreme gradient boosted decision tree regressor to the set of tested model. Jupyter notebook that trains the models and predicts are per lipids is available at `scripts/APL-predictor.ipynb` in the repository at https://github.com/NMRLipids/DataBankManuscript/. In further applications, it is important to consider potential limitations arising from the currently limited amount of data for certain types of systems. For example, the current models have been trained and tested only for mixtures with palmitoyl

and oleoyl acyl chains because the Databank does not yet contain enough data with varying number of double bonds. Such limitations are expected to alleviate with increasing amount of data in the future.

## Calculation of lipid flip-flops

Flip-flop rates were calculated using the `AssignLeaflets` and `FlipFlop` tools from the LiPyphilic package[68]. Headgroup atoms of each molecule, as defined in the mapping file, were used to determine in which leaflet the molecule locates. The midplane cut-off, defining the region between leaflets, was 1 nm and the frame cutoff was 100. This means that if the headgroup of a molecule entered within the distance of 1 nm from the bilayer midplane and was found in the opposing leaflet after 100 steps, this event was considered as a successful flip-flop event. The code that finds the flip-flop events from all simulations in the NMRlipids Databank is available at `scripts/FlipFlop.py`, and the results at `Data/Flipflops/` in the repository at https://github.com/NMRLipids/DataBankManuscript/.

## Analysing anisotropic diffusion of water in a membrane environment from NMRlipids Databank

Water permeability through membranes was calculated from equation $P = r/2c_w$, where $r$ is the rate of permeation events per time and area, and $c_w = 33.3679\,\text{nm}^{-3}$ is the concentration of water in bulk[37]. The number of permeation events in each trajectory was calculated using the code by ref. 38, available at https://github.com/crobertocamilo/MD-permeation. The code that calculates permeabilities for all simulations in the NMRlipids Databank is available at `/scripts/calcMD-PERMEATION.py`, and the resulting permeabilities are stored at `/Data/MD-PERMEATION` in the repository containing all analyses specific for this publication at https://github.com/NMRLipids/DataBankManuscript/. This repository is organized similarly to the NMRlipids Databank repository, enabling the upcycling of also the analyzed data without overloading the main NMRlipids Databank repository.

The lateral diffusion of water along the membrane surface, $D_{\parallel}$, was calculated with the Einstein's equation using the `-lateral` option in the `gmx msd` program within the Gromacs software package[69]. The code that calculates $D_{\parallel}$ for water from all simulations in the NMRlipids Databank is available at `/scripts/calcWATERdiffusion.py`, and the resulting diffusion coefficients are stored at `/Data/WATER-diffusion` in the repository at https://github.com/NMRLipids/DataBankManuscript/.

Water diffusion along the perpendicular direction of lipid bilayers in a multilamellar stack was estimated from the Tanner equation $D = \frac{D_{\parallel} P z_w}{D_{\parallel} + P z_w}$[44,45], where the water layer thickness, $z_w$, was estimated by subtracting the bilayer thickness from the size of the simulation box in the membrane normal direction.

## Reporting summary

Further information on research design is available in the Nature Portfolio Reporting Summary linked to this article.

# Data availability

All data related to this study are available via git repositories https://github.com/NMRlipids/Databank/(https://doi.org/10.5281/zenodo.7875567) and https://github.com/NMRLipids/DataBankManuscript/(https://doi.org/10.5281/zenodo.10252380). These repositories contain also source data files for all figures. Figures can be regenerated with codes mentioned in Methods section separately for each analysis. For instructions to access the data programmatically, see https://nmrlipids.github.io/.

# Code availability

All codes used in this study are available via git repositories https://github.com/NMRlipids/Databank/(https://doi.org/10.5281/zenodo.

7875567) and https://github.com/NMRLipids/DataBankManuscript/ (https://doi.org/10.5281/zenodo.10252380). For instructions to use the codes, see https://nmrlipids.github.io/.

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

## Acknowledgements

P.B. was supported by the Academy of Finland (Grants 311031 and 342908). F.F.-R. acknowledges Tecnológico Nacional de México, Dirección General de Asuntos del Personal Académico (DGAPA), Programa de Apoyo a Proyectos de Investigación e Innovación Tecnológica (PAPIIT) 101923, CONACyT Ciencia de Frontera 74884, for financial support and Miztli-Dirección de Cómputo y de Tecnologías de Información y Comunicación (DGTIC) - Universidad Nacional Autónoma de México (UNAM) (Project LANCAD-UNAM-DGTIC-057) facilities for computing time allocation. T.M.F. greatly acknowledges financial support by the Ministry of Economics, Science and Digitalisation of the State of Saxony-Anhalt. R.G.-F., A.P. and F.S.-L. thank Centro de Supercomputación de Galicia for computational support; R.G.-F. thanks Ministerio de Ciencia, Innovación y Universidades for a "Ramón y Cajal" contract (RYC-2016-20335), and Spanish Agencia Estatal de Investigación (AEI) and the ERDF (RTI2018-098795-A-I00, PDC2022-133402-I00), Xunta de Galicia and the ERDF (ED431F 2020/05, 02_IN606D_2022_2667887 and Centro singular de investigación de Galicia accreditation 2016-2019, ED431G/09); A.P. thanks Spanish Agencia Estatal de Investigación (AEI) and the ERDF (PID2019-111327GB-I00, PDC2022-

133402-I00), Xunta de Galicia and the ERDF (ED431B 2022/36, 02_IN606D_2022_2667887); F.S.-L. thanks Axencia Galega de Innovación for his predoctoral contract (02_IN606D_2022_266 7887), and Spanish Agencia Estatal de Investigación (AEI) and the ERDF (PID2019-111327GB-I00). N.K. was supported by the Slovak Scientific Grant Agency (VEGA 1/0223/20). M.K. acknowledges CSC — IT Center for Science for computational resources and thanks Finnish Cultural Foundation and the UEF Doctoral Programme for financial support. F.L. was supported by the Deutsche Forschungsgemeinschaft (DFG LO 2821/1–1). M.S.M. was supported by the Trond Mohn Foundation (BFS2017TMT01). L.M. acknowledges funding by the Institut National de la Santé et de la Recherche Médicale (INSERM). N.R. and R.T. acknowledge funding from Norges Forskningsråd (#288008 and #335772) and computational resources provided by Sigma2 - the National Infrastructure for High-Performance Computing and Data Storage in Norway. O.H.S.O, A.M.K, N.R. and L.S.B. acknowledge CSC — IT Center for Science for computational resources and Academy of Finland (grant nos. 315596, 319902 & 345631) for financial support. We acknowledge all the NMRlipids Project contributors for making development of the NMRlipids Databank possible.

## Author contributions

A.M.K. implemented the *Databank layer* and NMRlipids Databank-API. H.S.A. contributed to the quality evaluation metrics based on NMR and X-ray scattering form factors, wrote the machine learning code, and was involved in testing the GUI, polishing the manuscript, and in community outreach to source additional data. L.S.B. implemented roughly 630 pre-existing lipid bilayer simulations into the Databank and created the corresponding mapping files. P.B. implemented the PCA-based equilibration metric, supervised the work of A.K. and participated in code refactoring. F.F.-R. set up and performed simulations with Slipids Force Field: ID 617, ID 207 and ID 82. T.M.F. performed and analysed the solid-state NMR experiments. P.F.J.F. contributed to changes in the buildH software so that the Databank can handle united atom simulation data. R.G.-F. was responsible for the design, validation and supervision of the back-end and front-end for the GUI development (databank.nmrlipids.fi). I.G. advised on implementation of the PCA-based equilibration metric and supervised the work of A.K. B.K. contributed to the initial design and implementation of the Databank and was involved with writing the manuscript. N.K. contributed experimental scattering data. P.K. implemented the flip-flop analysis and supported the integration of PCA equilibration metric into the Databank. M.K. provided simulations with the OPLS3e force field. A.K. implemented the PCA-based equilibration metric. A.L. provided simulations of heterogeneous lipid membranes including sphingolipids and glycosphingolipids with the CHARMM36 force field. F.L. provided simulations using the CHARMM36m force field, incorporating cholesterol POPC and SAPI25 lipids. J.J.M. provided simulations with the Amber and CHARMM36m force fields, general discussion, and manuscript editing. M.S.M supported O.H.S.O. in conceptualising the project, contributed to the quality evaluation metrics based on NMR and X-ray scattering form factors, assisted in refining the manuscript and the GUI, and guided the work of L.S.B. and M.K. C.M. worked on the Databank implementation and contributed code improvements. L.M. performed simulations (for sphingomyelin, CHARMM36 force field, and Orange force field). N.R. implemented the code for the calculations of X-ray scattering form factors. A.M.N. provided simulations with TLCL lipid, simulations with Reaction-Field instead of PME, and improved the implementation. T.J.P. performed many of the all-atom and united-atom simulations and contributed to discussions on the project. A.P. was responsible for the design, validation and supervision of the back-end and front-end for the GUI development (databank.nmrlipids.fi). N.R. contributed simulations of complex lipid-bilayers obtained with NAMD and the CHARMM36 force field. S.S. contributed simulations of heterogeneous lipid membranes including sphingomyelin and ceramides with the CHARMM36 force field. F.S.-L. developed the back-end and participated in the design and development and validation of the GUI (databank.nmrlipids.fi). R.T. provided simulations of complex lipid-bilayers with the CHARMM36m force field. O.H.S.O. designed and supervised the work, implemented the water permeation, diffusion and machine learning analyses, and wrote the manuscript.

## Competing interests

The authors declare no competing interests.

## Additional information

[1]University of Helsinki, Institute of Biotechnology, Helsinki, Finland. [2]Department of Theory and Bio-Systems, Max Planck Institute of Colloids and Interfaces, 14424 Potsdam, Germany. [3]Department of Biomedicine, University of Bergen, 5020 Bergen, Norway. [4]University of Potsdam, Institute of Physics and Astronomy, 14476 Potsdam-Golm, Germany. [5]Nanoscience Center and Department of Chemistry, University of Jyväskylä, 40014 Jyväskylä, Finland. [6]Departamento de Ciencias Básicas, Tecnológico Nacional de México - ITS Zacatecas Occidente, Sombrerete 99102

Zacatecas, Mexico. [7]NMR group - Institute for Physics, Martin Luther University Halle-Wittenberg, 06120 Halle (Saale), Germany. [8]Sorbonne Université, Ecole Normale Supérieure, PSL University, CNRS, Laboratoire des Biomolécules (LBM), F-75005 Paris, France. [9]Université Paris Cité, F-75006 Paris, France. [10]Center for Research in Biological Chemistry and Molecular Materials (CiQUS), Universidade de Santiago de Compostela, E-15782 Santiago de Compostela, Spain. [11]Institute of Biological Information Processing: Structural Biochemistry (IBI-7), Forschungszentrum Jülich, 52428 Jülich, Germany. [12]ariadne.ai GmbH (Germany), Häusserstraße 3, 69115 Heidelberg, Germany. [13]Department of Physical Chemistry of Drugs, Faculty of Pharmacy, Comenius University Bratislava, 832 32 Bratislava, Slovakia. [14]Institute of Organic Chemistry and Biochemistry of the Czech Academy of Sciences, Flemingovo nám. 542/2, CZ-16610 Prague, Czech Republic. [15]School of Pharmacy, University of Eastern Finland, 70211 Kuopio, Finland. [16]Institut Charles Gerhardt Montpellier (UMR CNRS 5253), Université Montpellier, Place Eugène Bataillon, 34095 Montpellier, Cedex 05, France. [17]Heidelberg University Biochemistry Center, 69120 Heidelberg, Germany. [18]Department of Physics, University of Helsinki, FI-00014 Helsinki, Finland. [19]Department of Molecular Medicine, Morsani College of Medicine, University of South Florida, 33612 Tampa, FL, USA. [20]Center for Global Health and Infectious Diseases Research, Global and Planetary Health, College of Public Health, University of South Florida, 33612 Tampa, FL, USA. [21]Department of Chemistry, University of Bergen, 5007 Bergen, Norway. [22]Department of Informatics, Computational Biology Unit, University of Bergen, 5008 Bergen, Norway. [23]Hochschule Mannheim, University of Applied Sciences, 68163 Mannheim, Germany. [24]University of Lyon, CNRS, Molecular Microbiology and Structural Biochemistry (MMSB, UMR 5086), F-69007 Lyon, France. [25]Institut National de la Santé et de la Recherche Médicale (INSERM), Lyon, France. [26]Division of Pharmaceutical Biosciences, Faculty of Pharmacy, University of Helsinki, 00014 Helsinki, Finland. [27]Chemistry, University of Southampton, Highfield SO17 1BJ Southampton, UK. [28]Department of Applied Physics, Faculty of Physics, University of Santiago de Compostela, E-15782 Santiago de Compostela, Spain. [29]Institute of Biotechnology, RWTH Aachen University, Worringerweg 3, 52074 Aachen, Germany. [30]MD.USE Innovations S.L., Edificio Emprendia, 15782 Santiago de Compostela, Spain. [31]VTT Technical Research Centre of Finland, Espoo, Finland. ✉e-mail: samuli.ollila@helsinki.fi

