## [Peer Review File · Nature Communications]

Reviewers' comments:

Reviewer #1 (Remarks to the Author):

This manuscript presents a well-developed database of molecular simulation data for researchers to provide details of simple membrane models. This is important to the field of biophysics, but I do not find this revolutionary to justify publication to Nature Communications. This is more suited for a specialty journal based on my criticism below.

General Comments:

1. Refined database tool: This reviewer finds the database is important to the field of membrane biophysics and certainly could have more general applications. However, there lacks any novelty in this work that would capture the interest beyond those focused on simple model membranes that contain 1 or 2 lipids.

2. AI/ML: This manuscript starts with clear interest in using AI/ML and how this might be applied to lipid bilayers. However, this reviewer was expecting to see some development of an AI/ML tool set to take the database and predict properties of lipid membranes that lack experimental data. Yes, this database offers the ability to potentially develop such a tool but for this journal the actual tool that is developed would make this article more attractive to a general audience. As noted in the manuscript, membranes are difficult to study and those with multiple components even more so. If this manuscript presented an algorithm that learned how to predict membrane structure of bilayer with mixed lipids >3 components, then this would be more at the level for this high impact journal.

3. Water diffusion: It was nice to see how the database could be used for looking at water diffusion. The authors note that this is important for drug translocation in the skin. However, the lipid database lacks any skin lipids (ceramides) and the structure of skin is not bilayer-like. Moreover, there lacks any data in this database for drugs and thus the focus on water permeation lacks any general importance.

Reviewer #2 (Remarks to the Author):

This manuscript appears to have been submitted in haste. There are no authors, affiliations, Acknowledgements, Author contributions statement or Additional information.

The same is true for the database itself, which appears to only contain 741 trajectories and 452 membranes. On several pages links to PDB files are broken and images are missing.

The overall analysis held within the paper will likely be difficult to follow to a the more general audience of Nature Communications and I recommend a more specialist journal, either one focussing on the biophysical data or the database development.

I am sorry to not be more positive at this stage.

Reviewer #3 (Remarks to the Author):

Review:

Overlay databank unlocks data-driven analyses of biomolecules for all

The authors present the NMRlipids Databank, and demonstrate its application to find simulations that are in good agreement with experimental data (NMR order parameters, X-ray scattering). They also demonstrate that by analyzing many simulations in the databank, they may determine relationships between pairs of properties (order parameters vs. cholesterol flip-flops, water permeation vs. thickness, etc.). We can obtain an immense amount of information from analysis of a databank of MD simulations, and I certainly see this approach as being highly valuable.

The basic content of the databank is a number of parameters describing the setup of the MD simulations, plus online locations (Zenodo) of the simulations, such that one may first determine which simulations might be appropriate for analysis, and then download the simulations for further processing. The databank also contains some processed data that has been pre-analyzed by the authors (e.g. order parameters), and is stored along with the basic information on each trajectory.

I think the value of this work, however, depends on it providing a tool that one may somewhat easily be used to perform analyses over the whole databank or at least significant portions of it (“datadriven analyses of biomolecules for all”. This should allow the user to go beyond the parameters already calculated by the authors. Looking through the code for the Databank-API, I don’t think that

has been achieved, although there is a lot of potential for a powerful tool.

1) The API does not really have a user manual. There is information scattered in a blog (nmrlipids.blogspot.com) We do not know exactly to what the parameters in the databank (README.yaml files) refer. We also do not know exactly what functions are available for help in analyzing the databank.

2) There is very little commenting and limited help entries in the code in the API and in the included Jupyter notebooks. Furthermore, the Jupyter notebooks produce errors when run. It seems to be a simple problem of missing folders, but I think it is reasonable that functioning code be provided before publication.

3) There isn't really any adaptability in the scripts provided. For example, what if I want to compare my experimental data to the databank, instead of the provided experimental data. I don't see any way to do this with the plotQuality.ipynb script, since it appears the comparison is pre-calculated. So, then what script do we run to re-do the precalculation?

4) The databank makes information available to perform important operations, such as downloading and loading trajectories, and selecting the same atoms across different trajectories where the naming schemes are different (via mapping files), but there does not appear to be any accompanying functions to help utilize that information. For example, in the file 'calcOrderParameters.py', the following code has to be used to retrieve the trajectory. doi = system.get('DOI')

```
trj = system.get('TRJ')
trj_name = path + system.get('TRJ')[0][0]
trj_url = download_link(doi, trj[0][0])
#print(trj_name,tpr_name)
```

```
if (not os.path.isfile(trj_name)):
    response = urllib.request.urlretrieve(trj_url, trj_name)
```

Why not just have this built-in somewhere? For example, a system in the databank could have a function stored in its dictionary (or better yet, replace the dictionary with an object)

```
uni=system['load']() #Downloads the trajectory, loads it with MDAnalysis, returns MDAnalysis universe
```

```
uni=system.load() #Same idea, but make system an object (could be a dictionary-like object)
```

Similarly, there are mapping files to give a consistent naming scheme to the same molecule type from different force fields, but there are no functions to use them. For example, there ought to be a function that takes an MDAnalysis atom group ('ag') and returns a list of the standard names

```
std_names=ag2names(ag,system) #system is required to know which mapping file to use
```

Going the other direction, there should be a function to select by name

```
ag_final=select_by_name('M_G1_M',ag_initial,system) #maybe the name input should accept regex?
```

5) There is not any obvious structure to the API files. For example, the databank class is in the same file as the OrderParameter class, although additional files for order parameter calculation are found in the file "OrderParameter.py", and as far as I can tell, the actual calculation is run from the file "calcOrderParameters.py", which is itself in another folder entirely. Surely it would make some sense to reorganize these things so that someone unfamiliar with the project could easily follow the logic behind the code.

6) Some emphasis is placed on the separation of metadata and the actual simulations. I see the point— if I want to evaluate metadata on my laptop, this is no problem. On the other hand, it would be a major advantage to have the full data base (~1.9 TB) publicly available on a computational server somewhere, where analyses performed over the full databank could be run without waiting for downloads. NMRbox comes to mind, but I'm sure other options exist.

I'm excited about the databank, and I think our group could immediately start using it. However, if we did, we would write our own API. For publication of this paper, that's a big problem, because the authors make strong claims about the API:

"The NMRlipids Databank-API, available at github.com/NMRlipids/Databank, provides programmatic access to all

simulation data in the NMRlipids Databank through application programming interface (API). This enables wide range of

novel data-driven applications—from construction of machine learning models that predict membrane properties, to

automatic analysis of virtually any property across all simulations in the Databank.”

I don't find the above to be true. While the databank itself is quite useful and provides essential information for the simulations, the API is poorly documented, somewhat disorganized, and inconvenient to use. There needs to be a core set of tools that allow searching the databank, downloading and loading trajectories, and selecting specific atoms based on the universal naming scheme. These need to be well-commented and include standard help entries in the functions and classes. In a separated location, there needs to be scripts that are also well-commented, concise, and perform the analyses for basic parameters (order parameters, form factors, etc). It needs to be straightforward to use these scripts as templates for calculating user-defined parameters. Finally, in a third location, there need to be scripts (Jupyter notebooks would be nice), that run without errors and plot the various user defined parameters. These sort of exist already, but they aren't really commented and produce some errors. (So, three classes of code: basic tools, scripts for specific calculations that are stored in the databank, scripts for displaying/comparing results of the calculations). Right now, these are all in the same folder, with no naming distinction.

This work requires major revisions. The paper itself proposes exciting ideas on how to use many MD data sets in conjunction with experimental data to obtain dynamics information on lipids that is not obtainable based on one or a few simulations alone. On the other hand, the tools provided do not allow an external user to easily do what the paper claims (“data-driven analysis [...] for all”). I look forward to seeing updates to the API toolbox and well-documented/well-commented examples that make it possible to implement user-defined analyses.

Reviewer #1:

This manuscript presents a well-developed database of molecular simulation data for researchers to provide details of simple membrane models. This is important to the field of biophysics, but I do not find this revolutionary to justify publication to Nature Communications. This is more suited for a specialty journal based on my criticism below.

AUTHOR REPLY:

We thank reviewer for recognizing the quality and importance of our work. We have now substantially revised the manuscript to foster its impact beyond simple membranes. Particularly, we have followed the reviewer's suggestion and included machine-learning-based predictions of properties of various cellular membranes. We believe that these revisions make the manuscript more suitable for Nature Communications.

General Comments:

1. Refined database tool: This reviewer finds the database is important to the field of membrane biophysics and certainly could have more general applications. However, there lacks any novelty in this work that would capture the interest beyond those focused on simple model membranes that contain 1 or 2 lipids.

AUTHOR REPLY:

We now concretely demonstrate in the manuscript that the presented NMRlipids Databank enables construction of machine learning (ML) models, and that the developed models successfully predict properties of different multi-component membranes. The proof-of-principle ML models we have trained predict the lateral packing and thickness of complex membranes based on their lipid composition. Using these ML models, we now provide in Table 1 the predicted properties of membranes with lipid compositions corresponding various biological membranes. Lateral packing and thickness are important to many biologically-relevant membrane phenomena, such as permeation and cholesterol flip-flops as already demonstrated in the first version of manuscript, as well as membrane-protein interactions. Therefore, we believe that the prediction of these properties for membranes with complex lipid compositions (corresponding to different types of cellular membranes) provides novelty that captures the interest of wide audience studying complex membrane-containing systems in general.

2. AI/ML: This manuscript starts with clear interest in using AI/ML and how this might be applied to lipid bilayers. However, this reviewer was expecting to see some development of an AI/ML tool set to take the database and predict properties of lipid membranes that lack experimental data. Yes, this database offers the ability to potentially develop such a tool but for this journal the actual tool that is developed would make this article more attractive to a general audience. As noted in the manuscript, membranes are difficult to study and those with multiple components even more so. If this manuscript presented an algorithm that learned how to predict membrane structure of bilayer with mixed lipids >3 components, then this would be more at the level for this high impact journal.

AUTHOR REPLY:

Implementation of AI/ML models is nowadays relatively straightforward with the available programming tools, if the training data is programmatically accessible. Bottleneck for constructing such models for membranes has been the lack of available large data sets with such access. The NMRLipids Databank proposes solution for this issue based on the *overlay databank* concept. This approach enables the construction of wide range of AI/ML models learning from the data that is made programmatically accessible for all in the NMRLipids Databank.

Due to the wide potential range of such applications, our original plan was to examine these in more detailed in our upcoming articles. Nevertheless, we agree with the reviewer that at least a minimum practical example would make the current manuscript substantially stronger. Therefore, we have now included a model that predicts the lateral packing and thickness of a membrane based on its composition. This model is trained based on the data available in the NMRLipids Databank. For details, results and discussion, see the new sections in the revised manuscript: *2.4.2 Using NMRLipids Databank as a training set for machine learning applications: Predicting multi-component membrane properties* and *4.12 Training machine learning model to predict properties of multi-component membranes*.

3. Water diffusion: It was nice to see how the database could be used for looking at water diffusion. The authors note that this is important for drug translocation in the skin. However, the lipid database lacks any skin lipids (ceramides) and the structure of skin is not bilayer-like. Moreover, there lacks any data in this database for drugs and thus the focus on water permeation lacks any general importance.

AUTHOR REPLY:

Indeed, skin is not a “bilayer like” in the sense that it is not a single bilayer or multiple bilayers oriented in the same direction. However, it does contain structures that locally resemble lipid bilayers that are organized in a complex way. The diffusion of hydrophilic molecules, such as water, along and through the bilayers are used as parameters in pharmacokinetic models where the complex organization is included with other parameters. Importance of diffusion-anisotropy of water-like molecules in such models is becoming increasingly recognized, see for example refs. 12,40-42.

Although it is not related to the main points of these statements, the databank actually already contains also ceramides. With increasing amount of data, it will become possible to make more specific predictions from the databank on drug behavior in skin or other biological material. As the first approximation for hydrophilic molecules, we analyze here how the water permeation and anisotropic diffusion depends on membrane properties. Water diffusion is commonly analyzed in studies with similar goals, see for example ref. 48.

Importantly, the pharmacokinetic models are only one example of the importance of water dynamics in membrane systems. For example, potential applications in MRI imaging applications is mentioned in our manuscript (see ref. 11). Water diffusion through membranes is a highly active research topic with wide general interest from many points of view, see for example ref. 48. Therefore, we think that the water permeation analyzed in the manuscript bears, in fact, considerable general interest.

In the current version of the manuscript, we have revised the first two sentences of section 2.4.4. to underline that water anisotropic diffusion is relevant for models describing drug translocation through biological material: *“The anisotropic diffusion of water and hydrophilic molecules in directions parallel and perpendicular to membranes is an important parameter in models describing the translocation of drugs through biological material, particularly in the skin [12, 40–42]. Water anisotropic diffusion plays a role also in the signal formation in diffusion-tensor MRI imaging [11].”*

Reviewer #2 :

This manuscript appears to have been submitted in haste. There are no authors, affiliations, Acknowledgements, Author contributions statement or Additional information.

AUTHOR REPLY:

These sections were not included into the manuscript because we selected the double blind review option offered by the journal. For this reason, all the information related to author identities was removed from the version submitted to reviewers. Therefore, the lack of this information was due to journal policies, not a sign of haste.

REVIEWER:

The same is true for the database itself, which appears to only contain 741 trajectories and 452 membranes. On several pages links to PDB files are broken and images are missing.

AUTHOR REPLY:

The presented NMRlipids Databank is the largest publicly available collection of MD simulations with programmatic access. Furthermore, the databank is open for submissions and the amount of trajectories and membranes are steadily increasing in terms of numbers and diversity. In the manuscript, we demonstrate that already the current amount of data and tools delivered by the databank enable novel data-driven analyses of membranes with biological relevance that have not been possible before. Therefore, we do not fully understand the criticism concerning the number of trajectories and membranes in the databank.

Broken links to PDB files and missing images probably refer to the NMRlipids Databank-GUI (<https://databank.nmrlipids.fi/>)? However, we have not been able to reproduce such situations ourselves and it is therefore difficult to fix these without further information.

REVIEWER:

The overall analysis held within the paper will likely be difficult to follow to a the more general audience of Nature Communications and I recommend a more specialist journal, either one focussing on the biophysical data or the database development.

I am sorry to not be more positive at this stage.

AUTHOR REPLY:

We would like to stress that our manuscript does not just describe a regular database. We demonstrate how the *overlay databank* structure can be used to make MD simulations (or other data) programmatically accessible for all. As further demonstrated in the manuscript, this enables wide range of novel analyses of MD simulation data that have not been possible before. These results originally included analyses, for example, on cholesterol flip-flop and membrane permeation. Both of these are highly relevant biological phenomena that are widely discussed in the literature, see for example ref. 48 and Liu et al. [Nat. Chem. Biol. 13, 268-274 (2017)]. In the revised version, we also present a machine learning model that predicts membrane properties from its lipid composition, see the new section *2.4.2 Using NMRLipids Databank as a training set for machine learning applications: Predicting multi-component membrane properties*. With this model we predict the lateral packing and thicknesses of membranes with lipid compositions characteristic of different types of cellular membranes. We find these results to go far beyond just presenting a database, and to warrant the suitability of our manuscript for the general audience of Nature Communications.

Furthermore, we have now shortened and streamlined the discussion in the most technical part of Results and discussion section (related to the quality evaluation and selection of best simulation models). Also the panel describing “Result processing” in the application layer has been moved from Figure 1 into Figure S1 in the Supplementary information.

Reviewer #3:

The authors present the NMRLipids Databank, and demonstrate its application to find simulations that are in good agreement with experimental data (NMR order parameters, X-ray scattering). They also demonstrate that by analyzing many simulations in the databank, they may determine relationships between pairs of properties (order parameters vs. cholesterol flip-flops, water permeation vs. thickness, etc.). We can obtain an immense amount of information from analysis of a databank of MD simulations, and I certainly see this approach as being highly valuable.

The basic content of the databank is a number of parameters describing the setup of the MD simulations, plus online locations (Zenodo) of the simulations, such that one may first determine which simulations might be appropriate for analysis, and then download the simulations for further processing. The databank also contains some processed data that has been pre-analyzed by the authors (e.g. order parameters), and is stored along with the basic information on each trajectory.

AUTHOR REPLY:

We thank reviewer for the positive and constructive comments. We are especially grateful on extremely valuable feedback on the NMRLipids Databank-API (see below).

REVIEWER:

I think the value of this work, however, depends on it providing a tool that one may somewhat easily be used to perform analyses over the whole databank or

at least significant portions of it (“data-driven analyses of biomolecules for all”. This should allow the user to go beyond the parameters already calculated by the authors.

AUTHOR REPLY:

We fully agree with the reviewer on this. One of the main ideas in the design of the NMRlipids Databank is to enable flexible analyses of any parameters, that is, beyond the pre-calculated properties. Data for such analyses are made accessible for all by the *overlay databank* design and the universal nomenclature for atom and molecule names defined in the manuscript. We have now followed the highly valuable suggestions by the reviewer and substantially improved the documentation of the delivered tools for novel analyses. These improvements are detailed below.

REVIEWER:

Looking through the code for the Databank-API, I don't think that has been achieved, although there is a lot of potential for a powerful tool.

1) The API does not really have a user manual. There is information scattered in a blog (nmrlipids.blogspot.com) We do not know exactly to what the parameters in the databank (README.yaml files) refer. We also do not know exactly what functions are available for help in analyzing the databank.

AUTHOR REPLY:

The documentation of the NMRlipids Databank is now compiled in comprehensible format at: <https://nmrlipids.github.io/>. The user manual of the NMRlipids Databank-API can be found at: <https://nmrlipids.github.io/databankLibrary.html>, where also all the functions available to help analyzing the NMRlipids databank are listed and described. The parameters of README.yaml files were previously described in Table S3 in the manuscript and in GitHub; we have kept the Table S3 in the manuscript, but moved the description from GitHub to the NMRlipids Databank documentation: <https://nmrlipids.github.io/READMEcontent.html>.

REVIEWER:

2) There is very little commenting and limited help entries in the code in the API and in the included Jupyter notebooks. Furthermore, the Jupyter notebooks produce errors when run. It seems to be a simple problem of missing folders, but I think it is reasonable that functioning code be provided before publication.

AUTHOR REPLY:

Essential functions of the API have now been collected in `databankLibrary.py` and accompanied with docstrings. Documentation of this file is now available in the NMRlipids Databank documentation (<https://nmrlipids.github.io/databankLibrary.html>), along with the instructions on how to use the API.

In addition, there are now more thoroughly tested and better documented Jupyter notebooks available in a new repository: <https://github.com/NMRLipids/databank-template/tree/main/scripts>. These notebooks exemplify how to plot data from a selected simulation, how to show ranking tables of simulations against experiments, and how to

make user-defined analyses of simulations. These examples are now described also in the NMRlipids Databank documentation (<https://nmrlipids.github.io/exampleAndTutorials.html>). Overall, the approach of referring to files external to the notebook was improved such that there should not be errors due to folder references anymore.

REVIEWER:

3) There isn't really any adaptability in the scripts provided. For example, what if I want to compare my experimental data to the databank, instead of the provided experimental data. I don't see any way to do this with the plotQuality.ipynb script, since it appears the comparison is pre-calculated. So, then what script do we run to re-do the precalculation?

AUTHOR REPLY:

Instructions to add simulation or experimental data, and then update the databank are now added into the documentation, see <https://nmrlipids.github.io/addingData.html> and <https://nmrlipids.github.io/addingExpData.html>, respectively. Performing the instructed steps re-does the pre-calculation, and thereby enables also plotting the new user-contributed data. Support to perform these steps is also available from the active community through GitHub. Alternatively, it is possible to just contribute data, and let the administrators to run the databank update.

Furthermore, a new and improved templates to plot selected simulation data and to show rankings against experiments are added into the new template repository: <https://github.com/NMRLipids/databank-template/blob/main/scripts/plotSimulation.ipynb> and <https://github.com/NMRLipids/databank-template/blob/main/scripts/plotQuality.ipynb>, respectively.

REVIEWER:

4) The databank makes information available to perform important operations, such as downloading and loading trajectories, and selecting the same atoms across different trajectories where the naming schemes are different (via mapping files), but there does not appear to be any accompanying functions to help utilize that information. For example, in the file 'calcOrderParameters.py', the following code has to be used to retrieve the trajectory.

```
doi = system.get('DOI')
trj = system.get('TRJ')
trj_name = path + system.get('TRJ')[0][0]
trj_url = download_link(doi, trj[0][0])
#print(trj_name, tpr_name)
```

```
if (not os.path.isfile(trj_name)):
response = urllib.request.urlretrieve(trj_url, trj_name)
```

Why not just have this built-in somewhere? For example, a system in the databank could have a function stored in its dictionary (or better yet, replace the dictionary with an object)

```
uni=system['load']() #Downloads the trajectory, loads it with  
MDAnalysis, returns MDAnalysis universe  
uni=system.load() #Same idea, but make system an object (could  
be a dictionary-like object)
```

AUTHOR REPLY:

We have now implemented this in the *system2MDanalysisUniverse(system)* function. It takes the dictionary corresponding to a simulation as an input, and returns the MDAnalysis universe. This function also downloads the simulation files in the NMRlipids-Databank directory for further usage. This is now available in the NMRlipids Databank-API, see <https://nmrlipids.github.io/databankLibrary.html>.

We have opted to use functions rather than objects at this point, because these may be easier for users not comfortable with object-oriented programming.

Practical example of using this function can be now found in the new template for user-defined analyses (see the P-N vector calculation example):

<https://github.com/NMRLipids/databank-template/blob/main/scripts/template.ipynb>.

REVIEWER:

Similarly, there are mapping files to give a consistent naming scheme to the same molecule type from different force fields, but there are no functions to use them. For example, there ought to be a function that takes an MDAnalysis atom group ('ag') and returns a list of the standard names

```
std_names=ag2names(ag,system) #system is required to know  
which mapping file to use
```

AUTHOR REPLY:

This functionality is now implemented in *getUniversalAtomName(system, atomName, lipid)* function. It takes the system dictionary, simulation specific atom name, and universal lipid name as input. The function then returns the universal atom name corresponding to a given simulation-specific name and the defined lipid. This is documented in the NMRlipids Databank documentation:

<https://nmrlipids.github.io/databankLibrary.html#databankLibrary.getUniversalAtomName>

REVIEWER:

Going the other direction, there should be a function to select by name

```
ag_final=select_by_name('M_G1_M',ag_initial,system) #maybe the  
name input should accept regex?
```

AUTHOR REPLY:

This is now implemented in two functions: *simulation2universal_atomnames(system, molecule, atom)* and *read_mapping_file(mapping_file, atom1)*. The first one takes the

simulation system dictionary, molecule name, and atom name as an input. The latter takes the mapping file path and atom name as an input. The functions return the force field specific atom name. These are documented in the NMRlipids Databank documentation:

https://nmrlipids.github.io/databankLibrary.html#databankLibrary.simulation2universal_atomnames

REVIEWER:

5) There is not any obvious structure to the API files. For example, the databank class is in the same file as the OrderParameter class, although additional files for order parameter calculation are found in the file "OrderParameter.py", and as far as I can tell, the actual calculation is run from the file "calcOrderParameters.py", which is itself in another folder entirely. Surely it would make some sense to reorganize these things so that someone unfamiliar with the project could easily follow the logic behind the code.

AUTHOR REPLY:

We agree that there is room for improvement in software architecture of some programs in the NMRlipids Databank. Some of the analysis codes were written simultaneously with the development of the databank structure. This approach was selected to get immediate feedback on the functionality of the databank for automatic analyses. However, this slightly compromised the clarity of some codes because the core databank evolved simultaneously with the analysis code development. Now, when the core databank structure is established, many analysis codes could be clarified. Nevertheless, because these codes are functional, we have now concentrated rather to make new analysis codes, improve documentation and create new examples that not only enable applications of core databank but also further test its applicability and usefulness. We have now also slightly improved structure and documentation of, for example, the order parameter calculation code. Nevertheless, while we continue working on this, we think that software architecture and programming style is not a critical obstacle for publishing our manuscript.

REVIEWER:

6) Some emphasis is placed on the separation of metadata and the actual simulations. I see the point- if I want to evaluate metadata on my laptop, this is no problem. On the other hand, it would be a major advantage to have the full data base (~1.9 TB) publicly available on a computational server somewhere, where analyses performed over the full databank could be run without waiting for downloads. NMRbox comes to mind, but I'm sure other options exist.

AUTHOR REPLY:

We fully agree with the reviewer on the advantage of having the full databank on a publicly available server. However, one of the main principles guiding the design of the NMRlipids Databank is long-term stability, independent of individual researchers, institutes, or funding. This is now achieved with the *overlay databank* structure where data storage can be distributed to well-established resources, while the actual databank is stored as the git repository. Establishing a publicly available computational server without dedicated support from an institute or funding body is not straightforward; such server would require technical maintenance and a user-account

system, which would in turn require dedicated personnel for whom the long term support would be uncertain.

Nevertheless, the overlay design of the NMRlipids Databank does enable users to establish their own local or public servers. For example, a research group can download the databank, or parts of it, to a local server for their own usage. The access to such server can also be distributed more widely if possible and desired. Indeed, NMRbox could be one possible solution. We are currently actively investigating this and other possible solutions. However, we think that this should not be a critical barrier to publishing our manuscript.

REVIEWER:

I'm excited about the databank, and I think our group could immediately start using it. However, if we did, we would write our own API. For publication of this paper, that's a big problem, because the authors make strong claims about the API:

"The NMRlipids Databank-API, available at github.com/NMRlipids/Databank, provides programmatic access to all simulation data in the NMRlipids Databank through application programming interface (API). This enables wide range of novel data-driven applications—from construction of machine learning models that predict membrane properties, to automatic analysis of virtually any property across all simulations in the Databank."

I don't find the above to be true. While the databank itself is quite useful and provides essential information for the simulations, the API is poorly documented, somewhat disorganized, and inconvenient to use.

AUTHOR REPLY:

We have now organized the API better and written a more thorough documentation, as described in more detail above. Therefore we believe that the API is substantially more convenient to use now.

While we fully recognize the importance of the clarity and usability of the API emphasized by the reviewer, it should be noted that the main innovation in the manuscript is the *overlay databank* structure that enables many types of novel applications. While API is highly important, it is just one application in the "application layer" in the overlay databank structure. Due to the overlay structure and open collaboration approach, also alternative implementations of APIs are possible to make for everyone, for example, by the group of reviewer as suggested. Because such applications are now first time enabled by the NMRlipids Databank, we believe that the best practices to build APIs for this type of databank are yet to be found. The NMRlipids project community will continue this development, but we believe that imperfections in the current implementation of the API should not be a critical barrier to publishing our manuscript.

REVIEWER:

There needs to be a core set of tools that allow searching the databank, downloading and loading trajectories, and selecting specific atoms based on the universal naming scheme. These need to be well-commented and include

standard help entries in the functions and classes. In a separated location, there needs to be scripts that are also well-commented, concise, and perform the analyses for basic parameters (order parameters, form factors, etc). It needs to be straightforward to use these scripts as templates for calculating user-defined parameters. Finally, in a third location, there need to be scripts (Jupyter notebooks would be nice), that run without errors and plot the various user defined parameters. These sort of exist already, but they aren't really commented and produce some errors. (So, three classes of code: basic tools, scripts for specific calculations that are stored in the databank, scripts for displaying/comparing results of the calculations). Right now, these are all in the same folder, with no naming distinction.

AUTHOR REPLY:

The core set of tools that perform the functions mentioned by the reviewer are now collected in databankLibrary.py and documented in the NMRlipids Databank documentation: <https://nmrlipids.github.io/databankLibrary.html>. For more details, see also replies above.

Codes performing the basic analyses (order parameters, form factors, etc) still locate in <https://github.com/NMRLipids/Databank/tree/main/Scripts/AnalyzeDatabank> as previously. These are now simplified, and their running instructions are added in the NMRlipids Databank documentation: <https://nmrlipids.github.io/addingData.html>.

Example codes and templates to use the NMRlipids Databank are now available in a new repository: <https://github.com/NMRLipids/databank-template/tree/main/scripts>. The Jupyter notebooks are well commented and should run without errors. Examples to plot data from a selected simulation, to show ranking tables of simulations against experiments, and to make user-defined analyses of simulations are available. The new databank-template serves as a template also for advanced user-defined analysis from the NMRlipids Databank: <https://github.com/NMRLipids/databank-template/blob/main/scripts/template.ipynb>. These templates are also mentioned in the NMRlipids Databank documentation: <https://nmrlipids.github.io/exampleAndTutorials.html>.

We believe that the organization of the NMRlipids Databank and its documentation is now substantially clarified.

REVIEWER:

This work requires major revisions. The paper itself proposes exciting ideas on how to use many MD data sets in conjunction with experimental data to obtain dynamics information on lipids that is not obtainable based on one or a few simulations alone. On the other hand, the tools provided do not allow an external user to easily do what the paper claims (“data-driven analysis [...] for all”). I look forward to seeing updates to the API toolbox and well-documented/well-commented examples that make it possible to implement user-defined analyses.

AUTHOR REPLY:

We thank reviewer again for recognizing the potential of our manuscript, and particularly for delivering detailed, constructive, and highly valuable comments concerning the implementation of the NMRlipids Databank-API. We have now provided

NMRLipids Databank documentation that is available at <https://nmrlipids.github.io/index.html>. We believe that this, together with the above-detailed updates in the API toolbox and templates, makes implementation of user-defined analyses substantially easier for all.

However, we would like to point out that the main innovation in the manuscript is the design of the open-access *overlay databank* structure and its implementation. This enables a wide range of novel applications for all, which are not limited to a single implementation of the API. It makes quality-evaluated MD simulations accessible to all scientists with or without expertise and resources to produce such data themselves. The API is not a necessary requirement to use that information. While we totally agree with the reviewer on the importance of the easily usable API and its documentation, the databank can be also accessed with the provided GUI or with traditional command line tools without a specific API. The databank structure and design also enable anyone to build their own API or any other desired tools in the application layer.

REVIEWER COMMENTS

Reviewer #1 (Remarks to the Author):

This revision is an improvement to the original submission and the authors have refined the manuscript to make the work more impactful. Overall, I am happy with the changes and believe that this is important work that is worthy of publishing in Nature Communications. However, I do have a comment below that should be addressed.

General Comment:

1. Predicting APL of a bilayer: Figure 3 summarizes how the test set (20% of the total available data) does in comparison to the literature APL. The various regression models (Fig S6) show some variability but all appear to be biased toward having points below the $x=y$ line (over prediction of the APL). Is there a reason for this prediction to be biased toward larger APL?

Reviewer #3 (Remarks to the Author):

Review

Overlay databank unlocks data-driven analyses of biomolecules for all

My initial review of this paper focused on usability of the API that is distributed with the databank. Updates to the online documentation have addressed most of my concerns, and additional functionality has been added to the API to simplify its use. The new API functions that I suggested are implemented and also demonstrated in a Jupyter Notebook.

One thing that has not been done is making the full database (with trajectories) and API available on a server somewhere. I do think the ability to quickly do large-scale analyses of many or all trajectories will be beneficial.

Let us, however, accept that this should not be a requirement for publication. I do think that it should be straightforward to use the API, even if the authors consider their main achievement to be overlay

database, and not the API. This is necessary for reaching a broad audience, and this is typically required of journals such as Nature Communications. The better documentation and additional code mostly achieves that. I would, still, push for some kind of online implementation of the software, where users could start using the software without requiring extensive setup. For example, one can use software such as AlphaFold in Jupyter notebooks without any local installation.

(see <https://colab.research.google.com/github/sokrypton/ColabFold/blob/main/AlphaFold2.ipynb>)

So why not do the same thing with the various Jupyter Notebooks for this project?

Any publicly available Jupyter Notebook on GitHub is automatically transferred to Google Colab by changing the “github.com” to “githubcolab.com” in the web address. We usually include a button in our public Jupyter Notebooks to do this. For example, insert the following HTML into a Markdown cell (adjust so it points to the correct notebook):

```

```

The second step is to include a few lines of code in the notebook that will set it up in Colab if that's where it's being run. Cloning the database and installing MDAnalysis should be all that's required:

```
import sys

if 'google.colab' in sys.modules:

    !pip3 install MDAnalysis

    !git clone https://github.com/NMRLipids/Databank.git

    databankPath = '/content/Databank'

else:

    databankPath = '../Databank/'
```

I think easy accessibility is important for a project like this (at least, if it is to be a broad audience). These are a few simple things to implement to improve the accessibility.

Reviewer #1 (Remarks to the Author):

This revision is an improvement to the original submission and the authors have refined the manuscript to make the work more impactful. Overall, I am happy with the changes and believe that this is important work that is worthy of publishing in Nature Communications. However, I do have a comment below that should be addressed.

AUTHOR REPLY:

We thank the reviewer again for positive and relevant comments. We now discuss about the bias in machine learning predictions toward larger areas in the caption of figure S6.

Reviewer #1:

General Comment:

1. Predicting APL of a bilayer: Figure 3 summarizes how the test set (20% of the total available data) does in comparison to the literature APL. The various regression models (Fig S6) show some variability but all appear to be biased toward having points below the $x=y$ line (over prediction of the APL). Is there a reason for this prediction to be biased toward larger APL?

AUTHOR REPLY:

The most appealing explanation for this would be relatively low area per lipids predicted by CHARMM36 parameters used in the literature when compared to many other models in the training set. However, because extensive studies would be required to firmly conclude this, and because the small bias in the best models is not critically relevant for the conclusions in the current manuscript, we have opted not to investigate this more carefully at this point.

Nevertheless, we now mention this with a short discussion in the caption of figure S6:

“Our machine learning models that are trained using simulations with wide range of force field parameters have general tendency to predict larger area per lipid values than reported from CHARMM36 simulations in the literature. One explanation could be lower area per lipids predicted by the CHARMM36 parameters when compared with many other force fields in the training set⁵, yet other explanations cannot be ruled out.”

Notably, figure 3 actually contains predictions using 80% of the data (training set), not 20% of the data (test set).

Reviewer #3 (Remarks to the Author):

Overlay databank unlocks data-driven analyses of biomolecules for all

My initial review of this paper focused on usability of the API that is distributed with the databank. Updates to the online documentation have addressed most of my concerns, and additional functionality has been added to the API to simplify its use. The new API functions that I suggested are implemented and also demonstrated in a Jupyter Notebook.

One thing that has not been done is making the full database (with trajectories) and API available on a server somewhere. I do think the ability to quickly do large-scale analyses of many or all trajectories will be beneficial.

AUTHOR REPLY:

We thank reviewer again for highly useful comments in the initial review and in here. We agree with the reviewer on the usefulness of having trajectories readily available on a server, and we continue seeking a sustainable solution for this. Contacting NMRbox was not fruitful, but further

utilization of Colab, as suggested by the reviewer, has potential to lead a good solution. We have now implemented Colab versions of essential Jupyter notebooks as suggested by the reviewer (see the next comment), and we will further continue our work to make the databank even more usable.

Reviewer #3

Let us, however, accept that this should not be a requirement for publication. I do think that it should be straightforward to use the API, even if the authors consider their main achievement to be overlay database, and not the API. This is necessary for reaching a broad audience, and this is typically required of journals such as Nature Communications. The better documentation and additional code mostly achieves that. I would, still, push for some kind of online implementation of the software, where users could start using the software without requiring extensive setup. For example, one can use software such as AlphaFold in Jupyter notebooks without any local installation.

(see <https://colab.research.google.com/github/sokrypton/ColabFold/blob/main/AlphaFold2.ipynb>)

So why not do the same thing with the various Jupyter Notebooks for this project?

Any publicly available Jupyter Notebook on GitHub is automatically transferred to Google Colab by changing the “github.com” to “githubcolab.com” in the web address. We usually include a button in our public Jupyter Notebooks to do this. For example, insert the following HTML into a Markdown cell (adjust so it points to the correct notebook):

<https://colab.research.google.com/assets/colab-badge.svg>">

The second step is to include a few lines of code in the notebook that will set it up in Colab if that's where it's being run. Cloning the database and installing MDAnalysis should be all that's required:

```
import sys
if 'google.colab' in sys.modules:
!pip3 install MDAnalysis
!git clone https://github.com/NMRLipids/Databank.git
databankPath = '/content/Databank'
else:
databankPath = '../Databank/'
```

I think easy accessibility is important for a project like this (at least, if it is to be a broad audience). These are a few simple things to implement to improve the accessibility.

AUTHOR REPLY:

We have followed the excellent suggestion and instructions by the reviewer and created Colab badges to the notebooks that are listed in the Examples and tutorials in the documentation (<https://nmrlipids.github.io/exampleAndTutorials.html>). Codes that plot simulation results and their qualities are converted also to Streamlit apps at <https://lolicato-nmrlipids-gui-app-qa2ffe.streamlit.app/>. These apps ease visualization of simulation results and their rankings. In addition, the notebook that plots statistics from the NMRLipids databank is also added into the Examples and tutorials in the documentation with the Colab badge. The Colab versions and Streamlit app now enable starting to use the databank without extensive setup, thereby making NMRLipids databank more accessible. We will continue our work to seek optimal solutions to make the databank even more accessible in the future.